# SMPD: A soil moisture-based precipitation downscaling method for high-resolution daily satellite precipitation estimation

Kunlong He[1,2], Wei Zhao[1, *], Luca Brocca[3], Pere Quintana-Seguí[4]

[1]Institute of Mountain Hazards and Environment, Chinese Academy of Sciences, Chengdu 610041, China

[2]School of Civil Engineering, Sun Yat-sen University, Guangzhou 510275, China;

[3]Research Institute for Geo-Hydrological Protection, National Research Council, Perugia, Italy

[4]Ebro Observatory (OE), Ramon Llull University – CSIC, Roquetes, Spain.

*Correspondence*: Wei Zhao (zhaow@imde.ac.cn)

**Abstract.** As a key component in the water and energy cycle, precipitation with high resolution and accuracy is of great significance for hydrological, meteorological, and ecological studies. However, current satellite-based precipitation products have a coarse spatial resolution (from 10 to 50 km) not meeting the needs of several applications (e.g., flash floods and landslides). The implementation of spatial downscaling methods can be a suitable approach to overcome this shortcoming. In this study, we developed a Soil Moisture-based Precipitation Downscaling (SMPD) method for spatially downscaling the Integrated Multi-satellitE Retrievals for GPM (IMERG) V06B daily precipitation product over a complex topographic and climatic area in southwestern Europe (Iberia Peninsula), in the period 2016-2018. By exploiting the soil water balance equation, high-resolution surface soil moisture (SSM) and Normalized Difference Vegetation Index (NDVI) products were used as auxiliary variables. The spatial resolution of the IMERG daily precipitation product was downscaled from 10 km to 1 km. An evaluation using 1027 rain gauge stations highlighted the good performance of the downscaled 1 km IMERG product compared to the original 10 km product, with a correlation coefficient of 0.61, root mean square error (RMSE) of 4.83 mm and a relative bias of 5%. Meanwhile, the 1 km downscaled results can also capture the typical temporal and spatial variation behaviors of precipitation in the study area during dry and wet seasons. Overall, the SMPD method greatly improves the spatial details of the original 10 km IMERG product with also a slight enhancement of accuracy. It shows good potential to be applied for the development of high-quality and high-resolution precipitation products in any region of interest.

**Keywords:** GPM; SMPD; surface soil moisture; spatial downscaling; daily precipitation

## 1 Introduction

Precipitation, as a key driving force of the global water cycle under climate change conditions, changes greatly in space and time and is among the key factors affecting the hydrology, water resources and ecosystem of a watershed (Salzmann, 2016; Spötl et al., 2021). Hence, accurate and reliable spatial-temporal precipitation estimates are critical for

the assessment and understanding of climate change, hydrology, climatology, and its impacts on the environment, ecosystem, and human society (Xia et al., 2015; Wehbe et al., 2020; Wei et al., 2020; Bezak et al., 2021; Ma et al., 2021; Yang and Huang, 2021).

The most common ground-based method for precipitation measurement relies on rain gauge observations. Although rain gauges can provide accurate observations and capture the temporal variability in precipitation within a certain radius, these measurements are known to be prone to spatial representativeness issues due to the high spatiotemporal heterogeneity of precipitation (Wehbe et al., 2017; Tang et al., 2018). With the development of meteorological satellites, remote sensing has become the main tool for estimating regional to global precipitation because of its wide spatial coverage and continuous observation periods. These series of satellites include the Global Precipitation Climatology Project (GPCP) (Huffman et al., 1997), the Tropical Rainfall Measuring Mission (TRMM) Multisatellite Precipitation Analysis (TMPA) (Huffman et al., 2007), the NOAA Climate Prediction Center (CPC) morphing technique (CMORPH) (Joyce et al., 2004), Precipitation Estimation from Remotely Sensed Information using Artificial Neural Networks (PERSIANN) (Sorooshian et al., 2000), Global Satellite Mapping of Precipitation (GSMaP) (Kubota et al., 2007), and Integrated Multisatellite Retrievals for Global Precipitation Measurement (GPM) (Hou et al., 2014). Although each product has its strengths in the capture of precipitation spatial patterns, there is a common issue, induced by its coarse spatial resolution (e.g., 0.1°-0.5°), greatly blocking the application of these products in hydrological and meteorological research at the local scale (Lin and Wang, 2011; Prakash et al., 2016; Chen et al., 2018).

To enhance the applications of current coarse-resolution precipitation products, a procedure that involves spatially downscaling these products to fine scales has become an important solution. In recent decades, many downscaling methods have been proposed with the use of different satellite precipitation products. There are two major categories of downscaling methods: statistical downscaling and dynamical downscaling (Maraun et al., 2010; Tang et al., 2016). Statistical downscaling methods are mainly conducted by building the explanatory ability of the precipitation spatial distribution with fine-scale predictors, including topographic, geographic, atmospheric and vegetation variables, with the use of traditional regression methods (Xu et al., 2015; Ma et al., 2019b; Mei et al., 2020), optimal interpolation techniques (Shen et al., 2014; Chao et al., 2018), multidata fusion (Rozante et al., 2020; Ma et al., 2021), spatial data mining algorithm (called cubist) (Ma et al., 2017b; Ma et al., 2017a), geographical ratio analysis (Duan and Bastiaanssen, 2013; Ma et al., 2019a) and machine learning algorithms (He et al., 2016; Baez-Villanueva et al., 2020; Min et al., 2020). Due to their convenience and efficiency, these approaches are dominant in precipitation spatial downscaling research (Abdollahipour et al., 2021). Comparatively, dynamical downscaling refers to the use of regional climate models driven by global climate model output or reanalysis data to generate regional precipitation information (Rockel, 2015), which requires more information on internal mechanisms related to complex physical processes of precipitation, such as

atmospheric, oceanic and surface information (Tang et al., 2016). Hence, spatial downscaling is achieved by modelling the conditional distribution of precipitation at a fine scale to characterize the spatial structure of precipitation (Haylock et al., 2006; Munsi et al., 2021).

Among the existing methods, due to the computational efficiency and the consideration of orography and vegetation in precipitation distribution, the statistical downscaling methods have been widely used in recent years. Most of them were conducted with the use of predictors, such as topographic and vegetation factors (Immerzeel et al., 2009; Jia et al., 2011; Jing et al., 2016a; Zeng et al., 2021). However, these predictors do not have physical connections with precipitation, they act as important environmental variables influencing precipitation distribution. Consequently, the lack of the physical background of this type method may introduce high uncertainty to the downscaled results. Comparatively, surface soil moisture (SSM) presents an obvious and strong physical connection with precipitation via their coupling and feedback processes (Seneviratne et al., 2010). As indicated by Brocca et al. (2014). Precipitation is the main driver of SSM temporal variability. A sudden increase may occur in SSM after a rainfall pulse over a period of time, followed by a smooth recession limb driven by evapotranspiration and drainage. This relationship can be well reflected by an example of the time series of precipitation and SSM from Dec 26 to 28, 2017 at station BRAGANCA, Portugal (Figure 1). A rapid increase in SSM occurs after these rainfall events. Then, the moisture condition gradually becomes drier when there is no further rainfall.

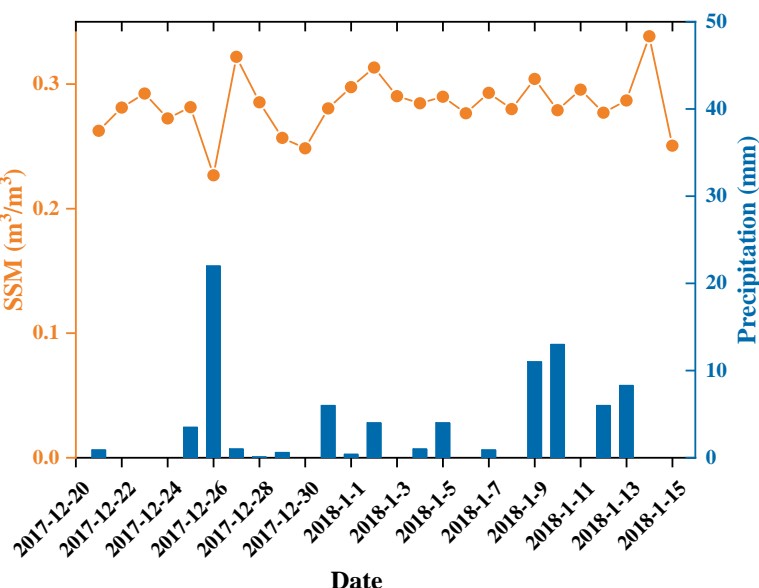

**Figure 1. Time series of observed precipitation and satellite observed SSM at station BRAGANCA, Portugal.**

According to this feature, SSM shows a big advantage in estimating precipitation, and this connection was approved by the SM2RAIN method proposed by Brocca et al. (2013). Fan et al. (2021) also demonstrated the good performance of the SM2RAIN products over the Tibet Plateau (TP) where the terrain is complex and the surface cover is heterogeneous. Additionally, the Soil Moisture Analysis Rainfall Tool (SMART) proposed by Chen et al. (2012) also

improved the sub-monthly scale accuracy of a multidecadal global daily rainfall product with a lower root mean square
error (-13%) and a higher probability of detection (+5%). Recent applications of this bottom-up approach further
demonstrate the success of using SSM in precipitation estimation at coarse-resolution scales (Brocca et al., 2016;
Ciabatta et al., 2017; Ciabatta et al., 2018; Brocca et al., 2019; Wehbe et al., 2020). Although there is a lagging effect of
the changes in soil moisture to precipitation, the rainfall-runoff experiment conducted by Song et al. (2020) further
confirmed this effect becomes small with the increase of the temporal aggregation interval and its impact is relatively
small at daily time scale (Brocca et al., 2016). Thus, it should be a very promising solution to improve the accuracy of
daily precipitation downscaling by introducing daily SSM in current downscaling schemes. However, the availability of
high-resolution SSM data is very limited and most of the current SSM products have a spatial resolution of more than
10 km (Peng et al., 2021), placing significant restrictions on these applications. Furthermore, suffering from an indirect
physical connection between topographic and vegetation factors and precipitation at a coarse temporal scale. Thus, a
large amount of downscaling research has been conducted at monthly or annual scales (Abdollahipour et al., 2021). In
addition, although daily high-resolution precipitation data have been produced by different methods (Brocca et al., 2019;
Hong et al., 2021), the use of high-resolution SSM data to improve the spatial resolution of satellite precipitation products
for generating daily-scale high-resolution precipitation data based on physical mechanisms is less studied.
In recent decades, there have been substantial progress in soil moisture downscaling studies (Merlin et al., 2008;
Piles et al., 2014; Peng et al., 2016; Tagesson et al., 2018; Long et al., 2019; Sabaghy et al., 2020; Wen et al., 2020; Zhao
et al., 2021), which makes the availability of high-resolution soil moisture data possible at a daily scale. Thus, the main
objective of this study is to establish a soil moisture-based precipitation downscaling (SMPD) scheme as a novel way of
obtaining fine-scale precipitation by fragmenting the coarse-pixel rainfall into fine-scale pixels. For this purpose, the 25-
km European Space Agency (ESA) Climate Change Initiative (CCI) SSM product is used to derive 1-km SSM data
based on the seamless downscaling method proposed by Zhao et al. (2021). Based on the inversion of the water balance
equation, a simplified model for estimating precipitation is constructed with the use of the downscaled 1-km seamless
soil moisture data and the vegetation index derived from the Moderate Resolution Imaging Spectroradiometer (MODIS)
observation and then applied to daily GPM precipitation products to obtain the daily downscaled precipitation estimates.
**2 Study area and datasets**
**2.1 Study area**
The central part of the Iberian Peninsula was selected as the study area (Figure 2). It is located in southwestern
Europe between 37.66°–42.99°N and 8.30° W–1.63° E. The region has a distinctly seasonal mild climate, with hot and
dry summers inland, cooler summers along the coast, and cold and wet winters. Precipitation presents a double peak
pattern, typical from the Mediterranean, with increased precipitation in Autumn and Spring. The central part of the study
area has a temperate continental climate, while the southern part has a Mediterranean climate, with warm and humid
winters and hot and dry summers. Generally, the south is dry and warm, while the north is relatively wet and cool.
Enhanced by the complex topographic pattern and diverse land cover conditions, this region has a highly heterogeneous
spatial environment, which makes this region a satisfactory candidate for precipitation downscaling. In addition, there
are many meteorological stations with long-term precipitation measurements in this area, which is an important
prerequisite for this study.

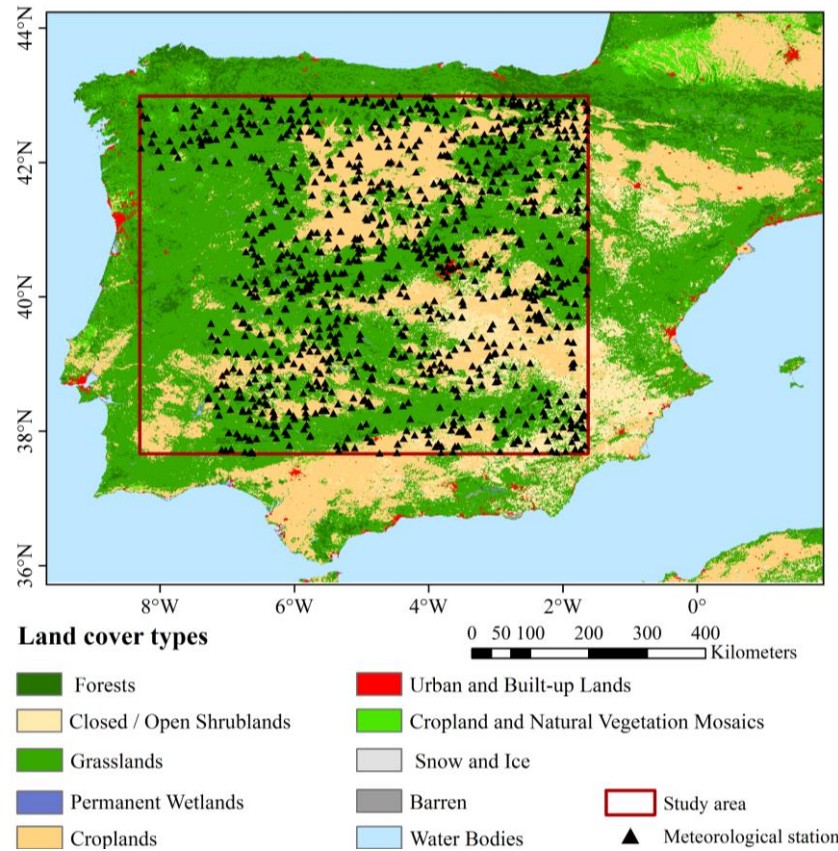

**Figure 2. Geolocation and land cover map of the study area. The black triangles denote the meteorological stations collected in this**
**study.**
**2.2 Datasets**
**2.2.1 GPM IMERG satellite precipitation data**
As the successor of the successful Tropical Rainfall Measuring Mission (TRMM), the Global Precipitation
Measurement (GPM) not only expands the measurement range and temporal and spatial resolution of the TRMM, but
also estimates the instantaneous precipitation more accurately, especially light-intensity precipitation (i.e., <0.5 mm h$^{-1}$)
and falling snow (Hou et al., 2014; Huffman et al., 2015), GPM-IMERG (Integrated Multisatellite Retrievals for GPM)
is the level 3 multisatellite precipitation algorithm of the GPM, which combines precipitation information measured from
the microwave sensor and infrared sensors onboard GPM constellations and monthly gauge precipitation data, and
IMERG employs the 2014 version of the Goddard Profiling Algorithm (GPROF2014) to compute precipitation estimates
from all passive microwave (PMW) sensors onboard GPM satellites, which is a significant improvement compared with
TMPA (GPROF2010) (Huffman et al., 2015; Huffman et al., 2020). Hence, it has attracted much attention in the satellite
remote sensing of precipitation.
Currently, the GPM product provides near-real-time products (early and late run) and postural-rime products (final
run) from sub-hourly to monthly resolution at a $0.1°×0.1°$ spatial scale. Owing to the infusion of multiple data, such as
microwave, infrared, radar, and Global Precipitation Climatology Centre (GPCC) rain gauge data (Hou et al., 2014), the
GPM-IMERG final run product provides more accurate estimates over the globe with a relatively long time series (June
2000- present) with a minimum latency of 3.5 months. In this study, the GPM-IMERG final run daily precipitation
product (downloaded from https://pmm.nasa.gov/data-access/downloads/gpm) was adopted as the downscaling object.
A three-year period from 2016 to 2018 was selected to verify the performance of the downscaling method based on the
availability of rain gauge data.

**2.2.2 ESA CCI surface soil moisture data**

The Soil Moisture CCI project is a part of ESA's Program on the Global Monitoring of Essential Climate Variables
(ECV), which was initiated in 2010 and has produced an updated SSM product annually since 1978 (Colliander et al.,
2017). The ESA CCI SSM series contains three separate SSM datasets, which are derived from active and passive
microwave remote missions as well as a combination of both, and the combined ESA CCI SSM product (version 04.7)
provides a spatial resolution of $0.25°$ and a temporal resolution of one day on a global scale (http://www.esa-
soilmoisture-cci.org/).
The combined ESA CCI SSM product provides the amount of water in the surface soil (approximately the top 5
cm), which integrates observations derived from 11 microwave sensors including active sensors such as Advanced
Scatterometer-A/B (ASCAT-A/B) and European Remote-sensing Satellite-1/2 (ERS-1/2), and passive sensors such as
Special Sensor Microwave Imager (SSM/I), the Scanning Multichannel Microwave Radiometer (SMMR), the TRMM
Microwave Imager (TMI), AMSR-E, WindSAT, AMSR2 and SMOS (Gruber et al., 2019). Previous evaluation studies
have demonstrated that ESA CCI SM generally agrees well with the spatial and temporal patterns estimated by land
surface models and in situ observations (Mcnally et al., 2016; Dorigo et al., 2017). Therefore, this combined product
was used in this study for the study period of January 1, 2016, to December 31, 2018, to obtain fine-resolution soil
moisture to assist in precipitation downscaling.

 **2.2.3 Normalized difference vegetation index (NDVI)**

NDVI is an important indicator of vegetation activity (Neinavaz et al., 2020; Zhang et al., 2020a; Pan et al., 2021),
especially for surface evapotranspiration (Joiner et al., 2018; Maselli et al., 2020). Therefore, it also presents a positive
correlation with precipitation (Quiroz et al., 2011; Birtwistle et al., 2016). The intuitive correlation between rainfall and
plant biomass represented by NDVI would enhance the downscaling study with high-resolution NDVI data. In this study,
the NDVI data were obtained from the MODIS/Terra 16-day vegetation index product
(https://lpdaac.usgs.gov/products/mod13a2v006/). It is a 16-day composite product obtained by choosing the best
available pixel value from all the acquisitions over 16 days with the spatial resolution of 1 km.
**2.2.4 Rain gauge data**
Daily precipitation data collected from 1027 rain gauge stations from 2016 to 2018 with different land cover
properties were used as the independent validation of the downscaled results in this study. These data were provided by
the Spanish State Meteorological Agency (AEMET). The distribution of the selected stations is mapped in Figure 2.
**3 Methodology**
**3.1 Soil moisture-based precipitation estimation model**
The soil water balance equation for a layer depth $Z$ can be described by the following expression:

$$Z\frac{ds(t)}{dt} = p(t) - g(t) - e(t) - r(t)$$

175                                                                                                          (1)

where $s(t)$ [-] is the relative saturation of the soil or relative SSM, $t$ is the time and $p(t)$, $r(t)$, $e(t)$ and $g(t)$ are the
precipitation, runoff, evapotranspiration, and drainage rate, respectively. By rearranging Eq. (1), precipitation can be
depicted as a function of SSM, runoff, evapotranspiration, and drainage rate. Based on this rule, Brocca et al. (2013)
proposed a bottom-up approach (SM2RAIN) by doing "hydrology backward" to infer precipitation with the use of
variations in SSM sensed by microwave satellite sensors. To perform this estimation, the model is simplified in different
ways by neglecting different components in Eq. (1) (Brocca et al., 2014; Massari et al., 2014) and the comparison study
indicated that the average contribution of surface runoff and evapotranspiration components amounts to less than 4% of
the total rainfall, while the soil moisture variation (63%) and subsurface drainage (30%) terms provide a much greater
contribution (Brocca et al., 2015). Although the contribution of evapotranspiration is relatively small, the dry
Mediterranean climate in most of this region emphasizes its importance. Therefore, the precipitation estimation model
was reorganized by only neglecting the runoff component:

$$p(t) = Z\frac{ds(t)}{dt} + g(t) + e(t)$$

187                                                                                                          (2)

In Eq. (2), the drainage rate is approximated by considering the relation in Famiglietti and Wood (1994) to include the
contribution of both deep percolation and subsurface runoff (interflow plus baseflow):
$$g(t) = as(t)^b \tag{3}$$
where $a$ and $b$ are two parameters expressing the nonlinearity between drainage rate and soil saturation. Regarding the
evapotranspiration component, there are many methods have been developed to estimate ET in natural ecosystems (Mu
et al., 2009; Sheffield et al., 2009; Carpintero et al., 2020). For instance, the daily evapotranspiration can be derived as
a function of the vegetation index ($VI$) and air temperature ($T_a$) (Nagler et al., 2005a; Nagler et al., 2005b):
$$e(t) = a\left(1 - e^{-bVI}\right)\left(m\big/\left(1 + e^{-(T_a - d)/p}\right) + f\right) \tag{4}$$
where the coefficients ($a, b, m, d, p,$ and $f$) were determined by conducting regression between ET and the independent
variables. Although there is a variable representing air temperature in Eq. (4) to specify the impact of air temperature
difference within a wide range, this variable can be assumed to be invariant when considering the pixels to a small extent.
Therefore, the term with the second brackets of Eq. 4 is simplified to the coefficient c, and Eq. (4) is further rewritten as
follows by introducing NDVI to present the $VI$ variable:
$$e(t) = c\left(1 - e^{-kNDVI}\right) \tag{5}$$
Based on the above approximation, the soil moisture-based precipitation estimation model was finally expressed
by the following equation:
$$p(t) = Z\frac{ds(t)}{dt} + as(t)^b + c\left(1 - e^{-kNDVI}\right) \tag{6}$$
where $ds(t)/dt$ can be calculated as the difference between the SSM estimates on nearby time steps. According to the
simplification in Eq. (6), this proposed model is appropriate for estimation to a local extent.
**3.2 Soil moisture-based precipitation downscaling (SMPD) method**
To perform precipitation downscaling, an important prerequisite is an assumption of spatial invariancy in the
precipitation estimation model described in Eq. (6) at coarse and fine scales, which is also the basis of many related
downscaling studies aiming at other surface parameters, such as soil moisture and temperature (Hutengs and Vohland,
2016; Mishra et al., 2018; Zhao et al., 2018; Ebrahimy and Azadbakht, 2019). Therefore, the estimation model
established at the 10-km level is thought to be applicable at the 1-km level. The estimated parameters $Z$, $a$, $b$, $c$ and $k$ at
10 km resolution scale resolution are not scale-independent, which can be used for the corresponding sub-pixel units (1
km). Moreover, because the downscaled model was constructed by using self-adaptive windows in different local regions
on the daily scale, these parameters vary in time and space. Thus, they are also temporal independent. The fitted
estimation model at 10 km scale was applied to the SSM and NDVI data at 1 km scale to obtain the estimated high-

resolution precipitation. Then, to preserve the mean rain rate over each coarse-scale pixel, the bias was corrected by redistributing the residual to each fine-scale pixel based on the kriging interpolation method. Finally, the downscaled daily GPM precipitation products were obtained with the integration of the estimated precipitation and the interpolated residual. According to the above principle, the downscaling method consists of the following parts and the main procedures in the downscaling processes are shown in Figure 3.

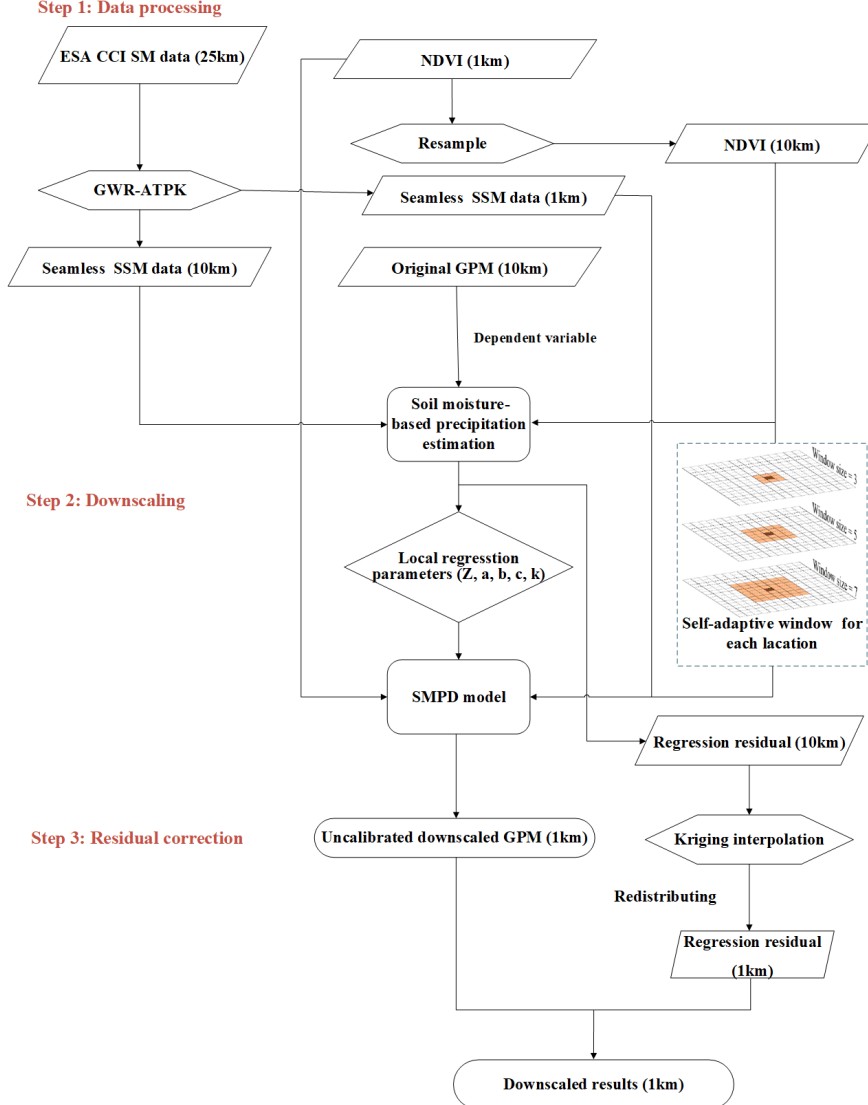

Figure 3. Flowchart of the process for downscaling the GPM data from 2016 to 2018.

### 3.2.1 Generation of daily SSM at a fine resolution

As shown in Eq. (6), SSM is an important variable in the estimation model. The ESA CCI SSM product can only provide coarse-resolution SSM data with unexpected gaps. To obtain daily SSM at a 1-km resolution, the seamless SSM downscaling method proposed by Zhao et al. (2021) is a good choice to achieve this goal. In comparison to the REMEDHUS network, the downscaled SSM performs better in terms of spatiotemporal coverage and evaluation metrics, which indicated that this method could be successfully used to produce high-resolution SSM data with no spatiotemporal gaps. This downscaling method mainly includes three steps: 1) filling gaps in the 25-km ESA CCI SSM maps with

neighbourhood information based on a local linear regression method, 2) estimating the 1-km regression SSM and
coarse-resolution residual with a geographically weighted regression (GWR) method, and 3) downscaling the coarse-
resolution residual to 1-km spatial resolution with the area-to-point kriging (ATPK) method and obtaining the fine-
resolution SSM. For details about the downscaling method, please refer to Zhao et al. (2021).
**3.2.2 Calibration of the precipitation estimation model with an adaptive window method**
Before model calibration, the 1-km downscaled SSM data and the NDVI data were first aggregated into a 10-km
scale to spatially match the spatial resolution of the GPM-IMERG product. Then, these data were applied to calibrate
the coefficients of the precipitation estimation model. As introduced in section 3.1, the application of this model requires
a prerequisite to work at a local extent because of the simplification of the evapotranspiration estimation. Therefore, a
local window with a radius from 3 to 7 cells was adopted in the fitting process. Initialized from the size of 3 cells, the
optimal window size was adaptively selected when the correlation coefficient (CC) of the fitting result reached the
maximum value. This adaptive method was applied to each coarse-resolution pixel with a sliding window, and the model
coefficients of this pixel were derived. During the model calibration, coarse pixels with zero precipitation were excluded.
$$p^{m}_{10\text{km}}(t) = Z\left(SSM_{10\text{km}}(t) - SSM_{10\text{km}}(t-1)\right) + aSSM_{10\text{km}}(t)^{b} + c\left(1 - e^{-kNDVI_{10\text{km}}}\right) \tag{7}$$

**3.2.3 Residual correction and fine-scale precipitation estimation**
Based on the calibrated estimation model coefficients in Eq. 7, the precipitation estimates determined with this
model can be calculated for each high-resolution pixel within the corresponding coarse pixel:
$$p^{m}_{1\text{km}}(t) = Z\left(SSM_{1\text{km}}(t) - SSM_{1\text{km}}(t-1)\right) + aSSM_{1\text{km}}(t)^{b} + c\left(1 - e^{-kNDVI_{1\text{km}}}\right) \tag{8}$$

However, there is a residual between the original precipitation value of each coarse-resolution cell pixel and the
mean value of the estimated precipitation of all fine-resolution pixels within this cell. For each coarse-resolution cell,
the residual is expressed as follows:
$$R_{10\text{km}} = p^{o}_{10\text{km}} - p^{m}_{10\text{km}} \tag{9}$$

The kriging interpolation method was used here to interpolate residuals $R_{10\text{km}}$ at coarse-resolution cells to obtain
kriging residuals fine-resolution scale (Wackernagel, 2003). The high-resolution residual was expressed as a weighted
integration of the residuals of the neighbouring coarse-resolution cells.
To meet the requirement of value preservation in the downscaling process, the kriging residuals should be corrected
by redistributing it to each fine-resolution pixel $i$. That is, the ratio of the $i^{th}$ high-resolution residual pixel in the $j^{th}$
coarse-resolution cell to the sum of the precipitation in the $j^{th}$ coarse pixel is used as the weight $\lambda_{ij}$, and the residual $R_{10\text{km}}$
is multiplied by the $\lambda_{ij}$, the kriging residuals were redistributed to each fine resolution pixel $i$ to obtain the residual after
value preservation can be expressed as follows:
$$R_{1km,ij} = \lambda_{ij} R_{10km,ij} , \text{ s. t. } \quad \lambda_{ij} = \frac{p^m_{1km, ij}}{\sum\limits_{i=1}^{n} p^m_{1km, ij}}$$  (10)
where $R_{1km,ij}$ represents the estimated precipitation of the $i^{th}$ high-resolution residual pixel in the coarse-resolution cell
$j$, $R_{10km,ij}$ represents the $j^{th}$ coarse-resolution cell residual in the self-adaptive window, $n$ is the number of high-resolution
residual pixels in the coarse-resolution cell, and $\lambda_{ij}$ is the weight coefficient of the $i^{th}$ high-resolution residual pixel in the
$j^{th}$ coarse-resolution cell. $p^m_{1km, ij}$ is the kriging interpolated residual $p^m_{1km, ij}$ at the fine-scale pixel $i$ in the $j^{th}$ coarse-
resolution cell.
Finally, the high-resolution precipitation was obtained by integrating the fine-resolution estimates via Eq. (8) and
the residual term in Eq. (10):
$$p_{1km} = p^m_{1km} + R_{1km}$$  (11)
**3.3 Validation**
To better assess the performance of the proposed downscaling method, the downscaled GPM results were validated
by observations from the collected stations in the study area at both daily and monthly scales. The evaluation metrics
include the correlation coefficient (CC), root mean square error (RMSE), and the relative bias (BIAS). They are defined
as follows:
$$CC = \frac{\sum\limits_{i=1}^{n} (S_i - \overline{S})(P_i - \overline{P})}{\sqrt{\sum\limits_{i=1}^{n} (S_i - \overline{S})^2 (P_i - \overline{P})^2}}$$  (12)
$$RMSE = \sqrt{\frac{\sum\limits_{i=1}^{n} (S_i - P_i)^2}{n}}$$  (13)
$$BIAS = \frac{\sum\limits_{i=1}^{n} (S_i - P_i)}{\sum\limits_{i=1}^{n} P_i}$$  (14)
where $P_i$ and $S_i$ are the precipitation measured by the rain gauge and satellite precipitation, respectively. $i$ is the index of the
precipitation series. $\overline{P}$ is the mean value of all rain gauge observations, and $\overline{S}$ represents the mean value of the satellite
precipitation, and $n$ represents the sample number of precipitation pairs.

281        Additionally, three metrics reflecting the capability of capturing precipitation events were introduced in the

assessment: the probability of detection (POD), the false alarm ratio (FAR) and critical success index (CSI). The POD
refers to the ratio of rain occurrences correctly detected to the total number of observed events; the optimum score is 1.
The FAR refers to the proportion of the precipitation events that the satellite falsely detects and the rain gauges do not
recognize it, the optimum score is 0. The CSI represents the fraction of precipitation events correctly detected by satellites
to the total number of observed or detected rainfall events, the optimum score is 1. The definition of a rainfall
accumulation "event" is one-day rainfall accumulation in excess of a given threshold of 0.1 mm. These three terms are
depicted as below:
$$POD = \frac{H}{H + M} \qquad (15)$$
$$FAR = \frac{F}{H + F} \qquad (16)$$
$$CSI = \frac{H}{H + F + M} \qquad (17)$$
where $H$ indicates the precipitation events concurrently detected by rain gauges and satellites, $M$ indicates the
precipitation events detected by rain gauges but not detected by satellites, and $F$ indicates the precipitation events
detected by satellites but not detected by rain gauges.
**4 Results**
**4.1 Accuracy of the soil moisture-based precipitation estimation model**

297        Before the downscaling process, the performance of the soil moisture-based precipitation estimation model was

evaluated first based on the calibrated estimation model in Eq. 7. Figure 4 shows the maps of the mean value of the daily
CCs and RMSEs during the period of 2016–2018 and their standard deviation (STD) by comparing the precipitation
estimated with the proposed estimation model and the GPM precipitation product at 10 km scale. Most of the CC values
are above 0.70 with an average value of 0.71, and most of the RMSE values are within the range from 0.50 to 1.00 mm,
with an average value of 1.00 mm. These results indicate the good consistency and small error between the estimated
precipitation and the original precipitation product. Furthermore, in view of the STD map, it represents the variability in
CC and RMSE during the period. The CC-STD values are within the range from 0.18 to 0.28 with an average value of
0.23, most of the RMSE-STD values are concentrated in the range of 0.50 to 1.50 mm, and only a few are in the range
of more than 3 mm, with an overall mean of 1.39 mm. Combined with the frequency distributions of CC and CC-STD,
RMSE, and RMSE-STD, the proposed estimation model can generally capture the precipitation with soil moisture
variations and it has relatively stable performance. According to the fitting performance assessment with the original

 GPM product, the soil moisture-based precipitation estimation model has been approved to be able to capture the

variation of precipitation with acceptable accuracy.

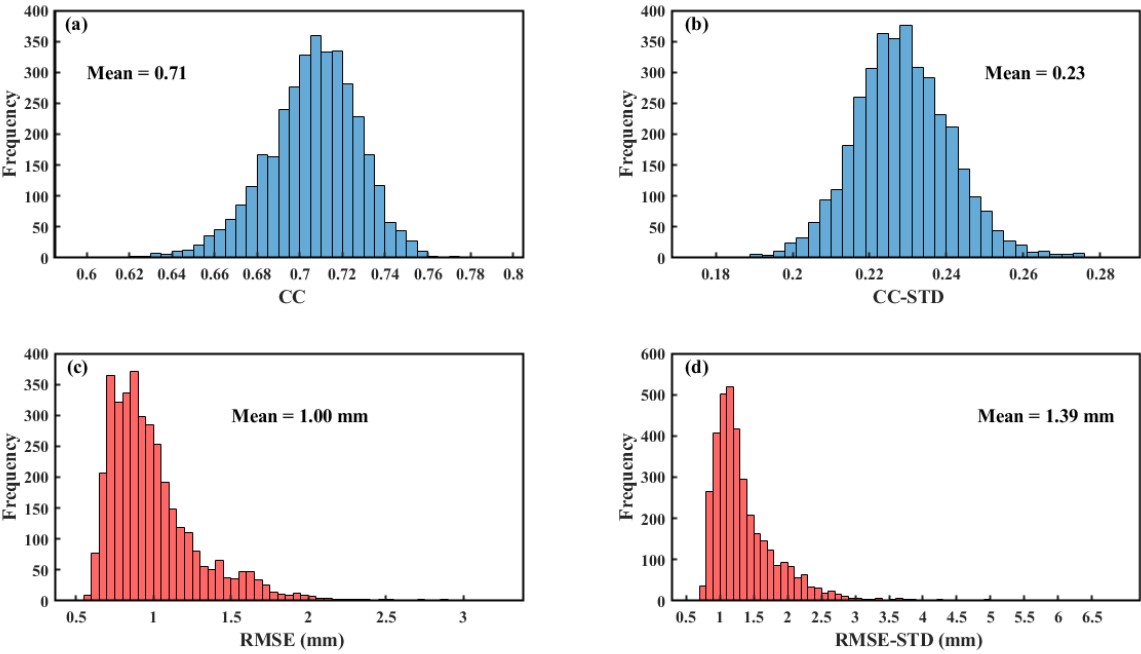


**Figure 4. (a) Maps of the mean value of the correlation coefficient (CC), (b) mean standard deviation of the CC (CC-STD), (c) mean root mean square error (RMSE), and (d) mean standard deviation of the RMSE (RMSE-STD) between the precipitation estimated with the soil moisture-based estimation model and the original GPM product during the period of 2016-2018. The mean value represents the average value of the corresponding index in the whole study area.**

**4.2 Overall performance of the downscaled precipitation**

**4.2.1 Spatial distribution**

To demonstrate the advantages of the downscaling results, two separate days (Jul. 7 and Nov. 25, 2017) in the dry season and wet season were selected to compare the original coarse-resolution precipitation data and the downscaled high-resolution precipitation data (Figure 5). From the visual inspection, the spatial distributions of the downscaled precipitation are highly consistent with those of the original ones in both seasons, especially for the distribution of the precipitation centers (>50 mm/day). The downscaled results maintained the original precipitation pattern in the GPM product, which can be reflected well by the very similar histograms of the original and downscaled precipitation on these two days, as shown in Figures 4c and f. In addition to their consistency, the downscaled results present higher spatial heterogeneity than the coarse-resolution product, which provides much more detailed information on the precipitation distribution within each coarse-resolution cell. More importantly, the downscaled results prevent the blockiness at the edges of the coarse-scale pixels.

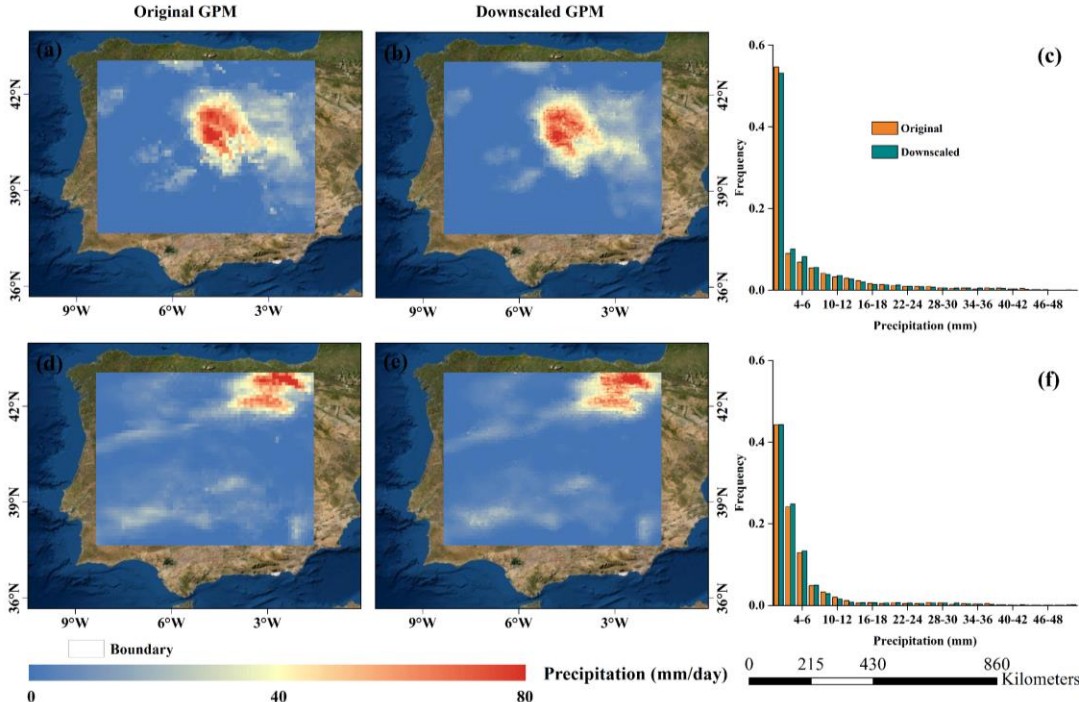

**Figure 5. Original daily GPM precipitation products, downscaled results, and their frequency histograms on July 7, 2017(a-c) and November 25, 2017(d-f).**

**4.2.2 Temporal variability**

In addition to the spatial distribution analysis, the temporal variation in the downscaled precipitation was further evaluated by introducing the downscaled results from Dec. 8 to Dec. 11, 2017. Figure 6 shows the daily maps of the original precipitation and downscaled precipitation. For the spatial distribution, both the original GPM precipitation product and the downscaled result have almost the same patterns on different days. Not only heavy rainfalls but also light rainfalls and no rains can also be captured by the proposed downscaling method in most circumstances. Moreover, the temporal variability in the daily precipitation was also preserved after the downscaling, and some outliers in the coarse-resolution GPM product were effectively filled with valid values, as shown by the downscaling results on Dec. 11 in Figure 6.

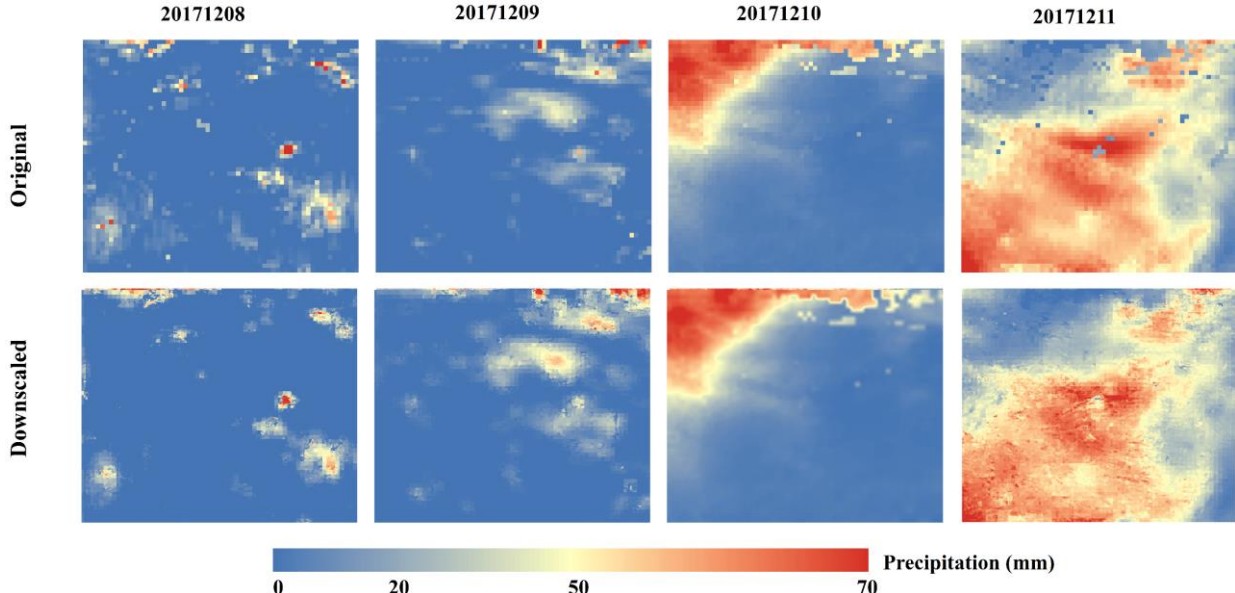

**Figure 6. Original daily GPM precipitation product and corresponding downscaled results from Dec.8th to Dec.11th, 2017.**

**4.3 Validation with rain gauge measurements**

**4.3.1 Validation at the daily scale**

To quantitatively evaluate the performances of the downscaling results, the daily original-scale GPM precipitation data and the downscaled results are compared separately with the precipitation measurements from all 1027 meteorological stations in the period of 2016 to 2018. Three metrics (POD, FAR, and CSI) for rainfall events, and CC, RMSE and BIAS for precipitation volumes, were used to make a comparison between the performances of both datasets. As shown by the density plots in Figure 7a, there is relatively high uncertainty in the original GPM precipitation product compared with the in-situ observation with a CC of 0.60, an RMSE of 4.99 mm and a BIAS of 9 %, which shows the GPM product generally overestimated observed precipitation at daily scale. These differences may be attributed to the differences in the spatial representativeness of both observations (one for the average value over a grid cell and one for a single point). Because of the value preservation during the downscaling process, the downscaled result also has a validation effect similar to that of the original GPM precipitation product (Figure 7b). However, compared with the original GPM product, the downscaled result shows an overall improvement in terms of CC, RMSE, and BIAS. There is a slight increase in CC, with its value increasing from 0.60 to 0.61. In contrast, both the RMSE and BIAS have a moderate reduction, with decreases of 0.16 mm and 4%, respectively. For rainfall event assessment, the downscaled result remarkably enhanced the ability to identify rainfall events at every station when compared with the original GPM product. Both the POD, FAR and CSI were moderately enhanced relative to those of the original GPM data, with an increasing POD from 0.84 to 0.88, a decrease in the FAR from 0.52 to 0.47 and an increasing CSI from 0.44 to 0.48. The comparison showed that the downscaled results could better detect precipitation occurrence than the original GPM product. The increase in spatial heterogeneity in the downscaled result assists rainfall event detection.

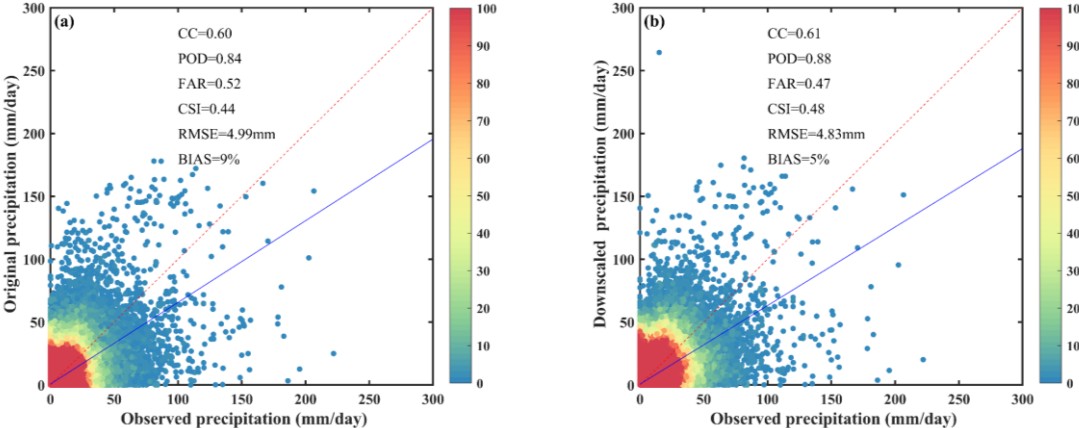


**Figure 7. Scatterplots of the original GPM precipitation product (a) and the downscaled results (b) plotted against daily precipitation**
**recorded by available meteorological stations over the study period. The red dotted line represents the 1:1 line and the blue solid**
**line represents the fitting line.**

366       In addition to the validation during the period of 2016-2018, further investigation was performed for the downscaled

results at individual months. Table 1 lists the evaluation indicators of the downscaled and original precipitation against
rain gauge observations for 1027 in-situ measurements from 2016 to 2018. In general, the downscaled results show
similar accuracy performance among different months from the detection accuracy of precipitation events reflected by
FAR and CSI. It is worth noting that the POD decreased compared to the original precipitation product, which may be
because compared with the coarse pixel precipitation at the daily scale, the downscaled products of the sub-pixels at the
same in-situ measurements location do not necessarily have precipitation, resulting in fewer precipitation events detected
by the downscaled products. From the RMSE values, seasonal differences can be detected. The dry season months from
June to September have relatively smaller RMSE values than other months. It is not because of the better performance
of the proposed method in these months but the inherent small precipitation of these months enables the low value of
RMSE. This feature can be also detected from the evaluation of the original data. Regarding the downscaled results
performance, the downscaled data have better accuracy in detecting precipitation events according to the improvement
in FAR and CSI in each month. Comparatively, the correlation feature of the downscaled results shows a small
improvement than the original data, represented by the CC values every month. Meanwhile, there are all decreasing
trends in terms of RMSE and the improvements in the wet seasons from October to May are relatively bigger than in the
dry season months. For the BIAS values, the improvements are also very clear with the extent from 3% to 7%. The
monthly comparison further indicated the improvement from the downscaled results which not only maintain the
temporal correlation characteristics of the original data with the gauge-based observations but also improve the absolute
accuracy according to the refinement of CC, POD, CSI, FAR, RMSE, and BIAS via introducing more detailed
information in the downscaling scheme.

**Table 1.** Validation of the downscaled precipitation data, and original GPM precipitation data with the daily precipitation measured by the selected stations at each month from 2016 to 2018.

| Month | Original | | | | | | Downscaled | | | | | |
|---|---|---|---|---|---|---|---|---|---|---|---|---|
| | CC | POD | FAR | CSI | RMSE (mm) | BIAS | CC | POD | FAR | CSI | RMSE (mm) | BIAS |
| January | 0.57 | 0.84 | 0.49 | 0.47 | 6.36 | 14% | 0.58 | 0.76 | 0.43 | 0.48 | 6.14 | 10% |
| February | 0.56 | 0.86 | 0.49 | 0.47 | 6.83 | 7% | 0.57 | 0.78 | 0.42 | 0.50 | 6.51 | 2% |
| March | 0.66 | 0.89 | 0.45 | 0.52 | 6.27 | -3% | 0.66 | 0.83 | 0.40 | 0.54 | 6.10 | -6% |
| April | 0.60 | 0.89 | 0.45 | 0.51 | 5.67 | 9% | 0.60 | 0.85 | 0.41 | 0.53 | 5.44 | 5% |
| May | 0.60 | 0.90 | 0.46 | 0.50 | 4.78 | 5% | 0.61 | 0.86 | 0.42 | 0.53 | 4.59 | 1% |
| June | 0.55 | 0.90 | 0.48 | 0.49 | 3.31 | 15% | 0.56 | 0.86 | 0.43 | 0.52 | 3.18 | 11% |
| July | 0.63 | 0.90 | 0.49 | 0.48 | 2.72 | 24% | 0.63 | 0.86 | 0.44 | 0.52 | 2.64 | 19% |
| August | 0.61 | 0.90 | 0.50 | 0.48 | 2.05 | 14% | 0.60 | 0.86 | 0.44 | 0.51 | 2.04 | 9% |
| September | 0.50 | 0.90 | 0.51 | 0.47 | 2.74 | 34% | 0.50 | 0.86 | 0.45 | 0.50 | 2.69 | 27% |
| October | 0.57 | 0.89 | 0.51 | 0.46 | 4.34 | 12% | 0.58 | 0.86 | 0.45 | 0.50 | 4.22 | 8% |
| November | 0.59 | 0.89 | 0.50 | 0.47 | 6.18 | 10% | 0.60 | 0.85 | 0.45 | 0.50 | 5.99 | 6% |
| December | 0.59 | 0.88 | 0.51 | 0.46 | 5.66 | 14% | 0.58 | 0.84 | 0.45 | 0.50 | 5.57 | 11% |

**4.3.2 Spatial distribution of the daily validation at all in-situ measurements**

In addition to the general evaluation with the measurements from all stations, the downscaled results are separately validated by the observations from each station, and the results are illustrated in Figure 8. In general, the downscaled precipitation estimates produce less error than the original GPM precipitation products with respect to all overall error statistics from 2016 to 2018, with an increase of CC values from 0.62 to 0.63, a decrease of RMSE values from 4.80 mm to 4.63 mm, a decrease of BIAS values from 17% to 13%, a decrease of FAR values from 0.50 to 0.45, an increase of POD values from 0.83 to 0.87 and an increase of CSI values from 0.47 to 0.50, respectively, which show moderate improvement compared to that of the original GPM products. Moreover, from the frequency histogram of validation indicators at 1027 in-situ measurements, the downscaled results present a better correlation with rain gauge observations with most of the CC values being above 0.71 in the central and north-western regions. Regarding RMSE values of downscaled results in Figure 8f, the validation at 728 in-situ measurements derives a low RMSE value (lower than 5.01 mm) and these stations are mainly located in the central and south-eastern regions. In comparison, the validation with high RMSE majorly occurred in the north-western regions due to the originally bigger annual mean precipitation. For BIAS, there is a relatively wide range from -72% to 99% in the whole region, systematic overestimation is observed at 685 stations, and underestimation is also observed at 342 stations. After downscaling, the overestimation was lightened. About the rainfall event assessment, most of the CSI values are higher than 0.48 at these stations and the FAR values are generally lower than 0.46, the POD values are generally higher than 0.81, as shown in Figure 8 j-r. It can also be seen that the detection accuracy of precipitation events in the humid northern region is better than that in the southern

region with less precipitation. Those results indicate that the fitting relationship between observed precipitation and
downscaled GPM products is good in the northwest region, while the errors in precipitation volumes are large in north-
western regions due to rich precipitation, which is consistent with the performance of the original GPM precipitation
product, while the accuracy was slightly better than that of the original precipitation product in the central and
southeastern regions. It proves that the improvement in rainfall events introduced by the downscaling method is not
limited to specific locations and covers the whole area, the downscaled results are more accurate in describing spatial
precipitation details.

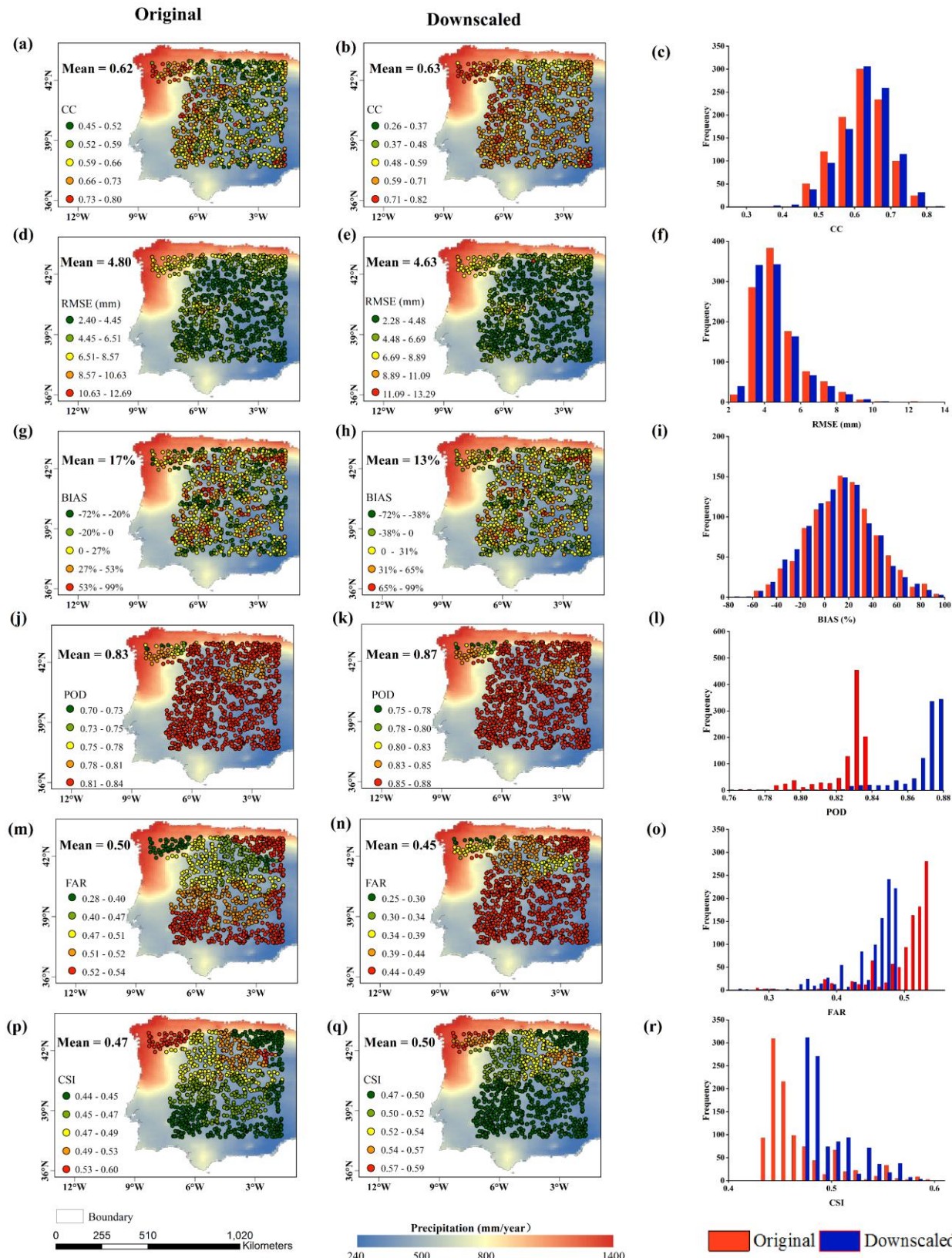


**Figure 8. CC (a-c), RMSE (d-f), BIAS (g-i), FAR (j-l), CSI (m-o) and corresponding frequency distributions for daily precipitation of original and downscaled GPM precipitation estimates at 1027 in-situ measurements during 2016–2018. The background value represents the original GPM annual average precipitation value from 2016 to 2018.**

Generally, the improvement from the overall performance for the downscaled results in Figure 8 is attributed to the number of improvements in the validation site indicators that occur between the original GPM product, the downscaled

results, and the observation stations at the daily scale. The downscaled results outperformed the original product in the detection accuracy of rainfall events and precipitation volumes, and the numbers of improvements in CSI and FAR are 1008 and 1026, respectively. Similarly, the number of improvements of CC, RMSE, and BIAS are 765, 886, and 884, respectively. The downscaled results are more accurate than the original product when they are validated by field measurements at most stations. In summary, the improvement in the precipitation downscaled by the SMPD method occurs at most rain gauge stations. The evaluation demonstrates the ability of this method to increase spatial heterogeneity to enhance the correlation with field measurements while also retaining the original GPM spatial distribution pattern. All the above results clearly prove the effectiveness of the downscaling method, which enhances daily GPM precipitation in both spatial information and accuracy.

### 4.3.3 Evaluation of precipitation intensities

To assess the downscaled GPM products' performance at different precipitation intensity intervals. The daily precipitation intensity is classified into five categories based on the rainfall thresholds (0, 10, 20, and 40 mm) Zambrano-Bigiarini et al. (2017). The performance metrics for the five daily precipitation intensity classes from 2016 to 2018 for 1027 in-situ measurements are listed in Table 2. In summary, original and downscaled GPM products performed the best in terms of all performance metrics for the no-rain events, while performing the worst for the violent rain events ($> 40$ mm $d^{-1}$). All precipitation products indicated that FAR values continuously performed the worst for the violent rain intensities, which showed that the products are still unable to accurately capture high precipitation values. Due to the reduced FAR values, the CSI value performed the best for no-rain events, followed by the light rain ([0, 10) mm $d^{-1}$), moderate rain ([10, 20) mm $d^{-1}$), heavy rain ([20, 40) mm $d^{-1}$) and violent rain events ($> 40$ mm $d^{-1}$), respectively. Additionally, the BIAS values showed that all precipitation products overestimated the number of light rain and underestimated moderate rain, heavy rain, and violent rain events. Most importantly, the performance of the downscaled precipitation product was slightly better than the original precipitation product for different rainfall intensity events in terms of CC, RMSE, POD, FAR and CSI values, indicating the reliability and accuracy of the downscaled products in capturing different rainfall intensity events than the original precipitation products.

**Table 2** CC, RMSE, BIAS, POD, FAR and CSI values for the different precipitation intensities for original and downscaled GPM products from 2016 to 2018 for 1027 rain gauge stations.

| Intensity (mm/d) | Original | | | | | | Downscaled | | | | | |
| | CC | RMSE (mm) | BIAS (%) | POD | FAR | CSI | CC | RMSE (mm) | BIAS (%) | POD | FAR | CSI |
| --- | --- | --- | --- | --- | --- | --- | --- | --- | --- | --- | --- | --- |
| 0 | - | 1.83 | - | 0.93 | 0.34 | 0.63 | - | 1.73 | - | 0.94 | 0.26 | 0.70 |

| 0-10 | 0.30 | 6.39 | 27.00 | 0.69 | 0.65 | 0.31 | 0.30 | 5.98 | 23.00 | 0.73 | 0.60 | 0.34 |
| 10-20 | 0.15 | 11.85 | -20.00 | 0.26 | 0.75 | 0.15 | 0.15 | 11.50 | -22.00 | 0.25 | 0.74 | 0.15 |
| 20-40 | 0.15 | 18.41 | -33.00 | 0.25 | 0.78 | 0.13 | 0.14 | 18.31 | -36.00 | 0.26 | 0.77 | 0.14 |
| >40 | 0.28 | 39.53 | -47.00 | 0.23 | 0.84 | 0.11 | 0.28 | 39.33 | -50.00 | 0.25 | 0.82 | 0.12 |

### 4.3.4 Validation at the monthly scale

In addition to the validation at the daily scale, the downscaling results were further evaluated at the monthly scale by integrating the daily results into the monthly amount. Figure 8 shows the multiannual average maps of the monthly precipitation from 2016 to 2018, including the original GPM product and the downscaled results. Similar to the daily comparison, the monthly distributions of both datasets have quite similar patterns over different months. The northern part of the study area has more precipitation than the southern part. The downscaled results maintain the precipitation centers in each month and depict the distributions around the centers well. The downscaled results can provide more detailed information regarding spatial distribution.

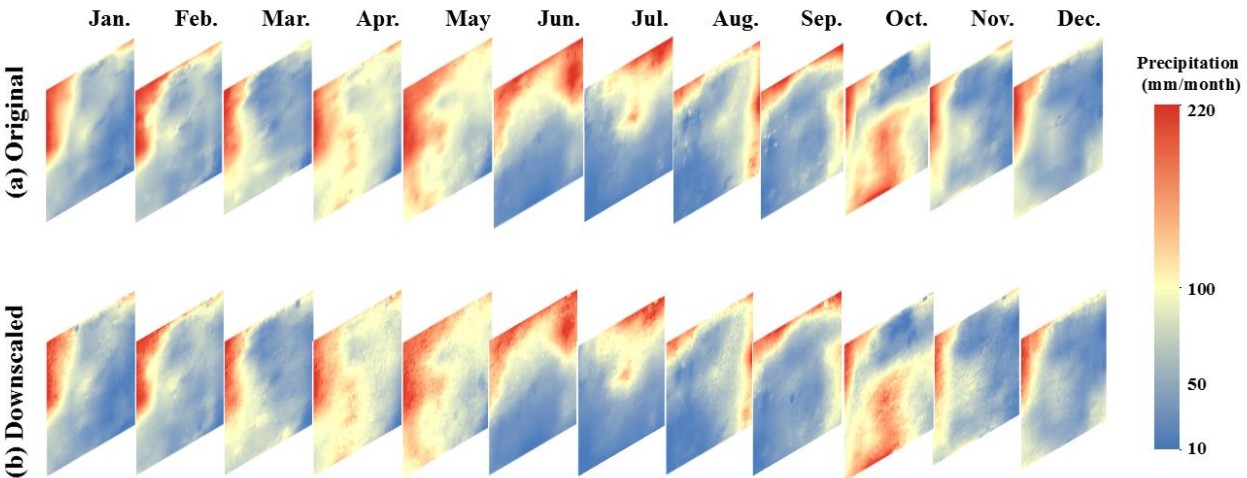

**Figure 9. Spatial distribution of the multiannual mean value of monthly precipitation for the original GPM product (first line) and the downscaled results (second line) from 2016 to 2018.**

By collecting the monthly precipitation of 1027 stations from 2016 to 2018, the accuracy of the monthly precipitation from the original and downscaled data was further quantitatively assessed. As shown in Figure 10a, after temporal integration, the uncertainty in the daily observation was greatly reduced in the monthly precipitation of the original GPM product. There is a significant increase in CC from 0.60 in Figure 6a to 0.83 in Figure 9a. However, systematic overestimation still occurs. After spatial downscaling, although there is no big change in terms of CC, both the RMSE and BIAS are clearly improved based on a comparison of the density plots in Figures 9a and b. For the analysis of the improvement ratio, only the performances of CC, RMSE, and BIAS are analyzed because the POD, FAR and CSI mainly reflect the rainfall events on the daily scale. Among the 1027 stations, the numbers of stations with improvements

during the validation in terms of CC, RMSE, and BIAS are 734, 587, and 912, respectively. Combined with the overall

validation and individual validation, the downscaled results at the monthly scale outperformed the original GPM product.

The evaluation shows that the downscaling method also presents good accuracy in the downscaling results and high

robustness at the monthly scale.

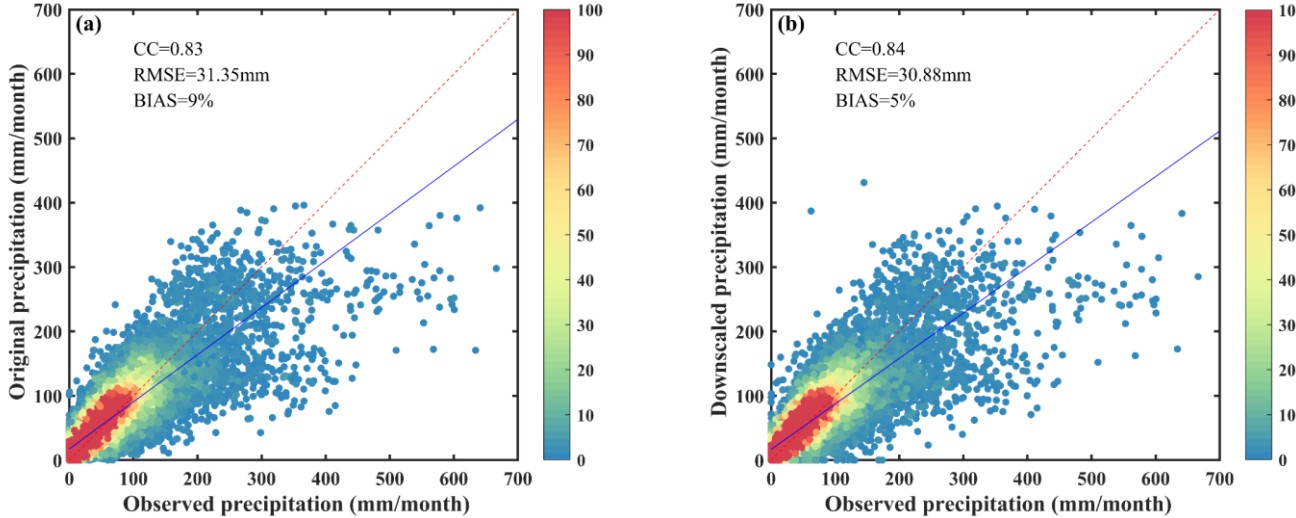

**Figure 10. Scatterplots of the original GPM precipitation product (a) and the downscaled precipitation data (b) plotted against the monthly precipitation measured by the meteorological stations during the period from 2016 to 2018.**

**5 Discussion**

In this study, a spatial downscaling method for coarse-resolution precipitation products was proposed to produce

high-spatial resolution precipitation data at a 1 km scale with the use of 1 km SSM data downscaled from microwave

remote sensing estimations. To establish the connection between SSM and precipitation, a simplified precipitation

estimation model based on the surface water balance equation was developed with inspiration from the SM2RAIN model

proposed by Brocca et al. (2014). By calibrating the model coefficients with a self-adaptive window at the coarse-

resolution scale, the precipitation model was applied to high-resolution variables to obtain the high-resolution estimates.

Compared with previous downscaling methods that mainly establish empirical relationships with surface variables, such

as NDVI and topographic factors, this method introduces the physical relationship between SSM and precipitation via

the water balance equation and has a solid physical basis. Therefore, the validation analysis conducted at both daily and

monthly scales indicated that the downscaled precipitation data outperformed the original precipitation product in most

circumstances and presented high robustness over three years with different rainfall strengthens.

**5.1 Advantages of the downscaling method**

In general, the SMPD method adopted the bottom-up approach in precipitation estimation, in which the variations

in SSM sensed by microwave satellite sensors have a strong connection with rainfall amounts according to the principle

of water balance (Brocca et al., 2014; Brocca et al., 2016; Mao et al., 2018). After a sudden increase in soil moisture induced by rainfall events, the moisture condition gradually becomes drier when there is no further rainfall. Therefore, this method has a clear physical mechanism and is the only downscaling method using SSM as the key driving factor. Comparatively, the traditional statistical downscaling methods were established based on the statistical relationship between environmental factors and precipitation. Take the spatial interpolation method as an example, although the application of this method is convenient, the accuracy of the interpolated precipitation data is limited by the rainfall gauge density, especially in the mountainous watershed with complex topography (Zhang et al., 2020b; Guo et al., 2021). The high dependency on in-situ measurements constrains its applications in the area with few observations. In contrast, the SMPD method breaks the limitation caused by the rainfall gauge density and has a broader application prospect.

To further demonstrate the advantage of the SMPD method, it is beneficial to compare the validation accuracy of this method with the validation accuracies of existing downscaled approaches, as shown in Table 3. In current existing downscaling studies, the involvement of daily SSM ensures downscaling at a daily scale is rarely considered. However, the relationship between SSM and precipitation ensures the daily downscaling in the proposed SMPD method. Comparatively, although Yan et al. (2021) conducted daily precipitation downscaling with the use of the random forest (RF) method, the RMSE value was considerably lower than that of the SMPD method. Moreover, this machine-learning method is highly dependent on the available training dataset. Comparatively, the daily or sub-daily downscaling studies conducted by Long et al. (2016) and Chao et al. (2018) have relatively better performances in terms of RMSE and CC, respectively. However, the incorporation of gauge precipitation data in the downscaling process partly enhances the estimation accuracy. These methods highly rely on in situ measurements without the independence of rain gauge measurements. In a recent hour-scale downscaling study conducted by Ma et al. (2020a), a geographically moving window weight disaggregation analysis (GMWWDA) method was developed by introducing cloud properties as covariates to downscale GPM precipitation products. Although it provided estimates at a very high temporal frequency, the limited rainfall-related environmental variables at the 0.01°/hourly scale constrained its application.

For the intercomparison of the monthly accuracy, the daily downscaled results of the proposed method outperformed most of the previous monthly downscaling studies using either RF or GWR algorithms (Jia et al., 2011; Xu et al., 2015; Jing et al., 2016b; Chen et al., 2018; Zhan et al., 2018). As shown in Figure 9b, the CC value was higher than most of them in the abovementioned studies. Although the RF-based downscaling method in Jing et al. (2016b) has a relatively low RMSE, the measurements from in situ stations were used to train the downscaling model which greatly reduces the dependence of the downscaling process on field observations. A similar requirement is also presented in Lu et al. (2019) and Long et al. (2016), and the GWR and multivariate regression models are largely dependent on the number of available training stations and variables related to the geophysical mechanisms of precipitation. The

independence of field observations in the SMPD method shows a large advantage, especially for regions with sparse
meteorological stations. Zeng et al. (2021) also proposed an independent downscaling approach considering temporal
lag from vegetation changes to precipitation. However, the relationship shows high variability which may result in a
negative correlation within a short time. Therefore, both the CC and RMSE of this method have worse performances
than those of the proposed method. In general, according to the methodology comparison, the proposed SMPD method
exhibits good performance in terms of both CC and RMSE. Unlike using the empirical regression method to build the
relationship between precipitation and other surface variables, the SMPD method demonstrated high effectiveness,
independence, and robustness.
**Table 3.** List of the performance of downscaling procedures to improve the spatial resolution of satellite precipitation products at different temporal scales. The bold
letters represent the proposed method in this study.

| Original products | Downscaled algorithm | Auxiliary variables | Temporal resolution | Downscaled products | | | Reference |
| --- | --- | --- | --- | --- | --- | --- | --- |
| | | | | Spatial resolution | CC | RMSE (mm) | |
| TRMM (25 km) | RF | DEM, NDVI | Monthly | 1 km | 0.86 | 15.70 | Jing et al. (2016b) |
| GPM (10 km) | GWR | DEM, NDVI | Monthly | 1 km | 0.79 | 20.94 | Lu et al. (2019) |
| GPM (10 km) | GWR | DEM, NDVI | Monthly | 1 km | 0.79 | 27.23 | Zhan et al. (2018) |
| TRMM (25 km) | GWR | DEM, Rain gauge data | Monthly | 1 km | 0.87 | 46.14 | Chen et al. (2018) |
| TRMM (25 km) | GWR | DEM, NDVI | Monthly | 1 km | 0.82 | 25.10 | Xu et al. (2015) |
| GPM (10 km) | RF | DEM, NDVI, LST | Daily | 1 km | 0.64 | 6.06 | Yan et al. (2021) |
| TRMM (25 km) | Multivariate regression model | DEM, Climate data | Daily | 1 km | - | 2.71 | Long et al. (2016) |
| GPM (10 km) | LPVIAL | NDVI | 16-day | 1 km | 0.81 | 46.77 | Zeng et al. (2021) |
| CMORPH (8 km) | GWR | DEM, NDVI | 30 min | 1 km | 0.86 | 7.27 | Chao et al. (2018) |
| GPM (10 km) | AMCN, GDA | LST, EVI, LSR | Monthly | 1 km | 0.83 | 30.88 | Jing et al. (2022) |
| GPM (10 km) | GMWWDA | Cloud Property Data | Hourly | 1 km | 0.53 | 5.16 | Ma et al. (2020a) |
| GPM (10 km) | SVM | Atmospheric, variables, DEM | Daily | 1 km | 0.78 | 12.55 | Min et al. (2020) |
| **GPM (10 km)** | **SMPD** | **SSM, NDVI** | **Daily** | **1 km** | **0.61** | **4.83** | **Proposed method** |


**5.2 Limitations and prospects**

Despite the superior performance of the SMPD method, some issues still need to be considered in practical applications. The first issue should relate to the accuracy of the original GPM precipitation data. Due to the limitation of the inherent accuracy of original GPM precipitation data, which are mainly manifested in two aspects, firstly the IMERG-Final products are corrected on a monthly scale using the interpolated precipitation product Global Precipitation Climatology Centre (GPCC, 1.0°/Monthly) based on ground observations. However, there is no mature calibration algorithm for calibrating the daily satellite-based precipitation estimates (Ma et al., 2020b). Second, the prior databases of cloud cover and precipitation profiles for retrieving passive microwave-based satellite precipitation estimates are not sufficiently robust due to the lack of ground-based radar observations. In addition, since passive microwave remote sensing-based precipitation retrieval is the primary input to the IMERG-Final products, it may lead to poor performance of the satellite-based product in winter and high-latitude regions (Xu et al., 2022). Therefore, the improvement in the accuracy of downscaling results is limited because of the value preservation during the downscaling process. The downscaling performance is highly dependent on the accuracy of the original GPM products. The multisource data fusion model based on observed rain gauge stations and reanalysis data proposed by Ma et al. (2021) and Li and Long (2020) could increase its ability to describe the daily precipitation fluctuations and it would help provide more accurate downscaling precipitation values. Given the spatial inconsistency of the point measurement and grid-scale estimation, which may lead to some uncertainty in the evaluation results. Thus, the difference in spatial scale between satellite and gauge-based precipitation measurements should be paid more attention to in future comparisons based on reanalysis-based precipitation with high spatial resolution.

In addition, the uncertainty of SSM and the sensitivity relationship between SSM and precipitation under continuous rainfall conditions may introduce uncertainty in the downscaling precipitation results. First, the responses of SSM with different land cover conditions and vegetation coverages to precipitation are relatively different (Fan et al., 2021), and topographic factors such as depressions and slopes also affect the uncertainty of SSM. Therefore, it is necessary to establish the relationship between SSM and precipitation for different land cover types or different terrain types. The establishment of a more reliable fitting relationship based on precipitation data with different land cover properties or topographic factors would be helpful to enhance the accuracy of the downscaling results (Chen et al., 2020; Senanayake et al., 2021; Zhao et al., 2021). Second, although the relationship between SSM and precipitation has been well demonstrated in many previous studies, the sensitivity of SSM to precipitation may decrease when soil water storage becomes saturated after repeated precipitation (Song et al., 2020). Therefore, it is necessary to further improve the relationship by considering the soil water threshold saturation in future studies. Moreover, this downscaling method was based on the surface water balance principle, and the runoff factor under heavy precipitation conditions at a certain time

was not considered because of the inherent scarcity of high-resolution runoff datasets from in situ measurements. Some
studies have provided good alternatives to obtain runoff data with high spatiotemporal resolution (Jadidoleslam et al.,
2019; Muelchi et al., 2021). Hence, the use of this runoff factor in the water balance equation for heavy precipitation
will assist in improving downscaling accuracy.
Most importantly, many previous studies have successfully generated fine precipitation data at hourly or half-hourly
scales (Ma et al., 2020a; Ma et al., 2020b; Lu et al., 2022; Ma et al., 2022). Nevertheless, these studies lacked physical
mechanisms in the downscaling process and do not use surface soil moisture covariates that respond in real-time to
precipitation. In the proposed method, the key inputs of the downscaling process are surface soil moisture and
precipitation data. Even on hourly or half-hourly scales, the soil moisture exhibits an instantaneous response to collocated
precipitation. Then, the soil moisture estimation method achieved seamless downscaling for high-resolution soil moisture
generation under cloudy conditions. Therefore, it would be able to obtain real-time soil moisture from microwave
satellite observations combined with surface temperature and vegetation index derived from optical and thermal infrared
remote sensing. Therefore, this approach has the potential for generating high spatial resolution precipitation data at
hourly or half-hourly scale.
**6 Conclusions**
In this paper, by introducing high-resolution SSM data and the NDVI as independent variables, a novel physical
downscaling approach based on the principle of surface water balance is developed to obtain high-resolution (1 km × 1
km) daily precipitation estimation. At both daily and monthly scales, the downscaled precipitation presents a similar
spatial and temporal distribution pattern as the original GPM product. Furthermore, a systematic evaluation of the
downscaled GPM data was conducted on multiple time scales at the station level. The downscaled precipitation showed
a good correlation with the observed measurements at each station at the daily scale, with POD, FAR, CSI, CC, RMSE,
and BIAS values of 0.88, 0.47, 0.48, 0.61, 4.83 mm, and 5%, respectively, and the evaluation results outperformed the
original GPM product. For monthly scale comparison, the downscaled data also presented a strong correlation with the
observed precipitation, with CC, RMSE, and BIAS values of 0.84, 30.88 mm, and 5%, respectively. With the increase
in spatial heterogeneity in the downscaled results, there is also an increasing trend in the improvements in the
precipitation accuracy through the comparison at most stations.
In summary, the proposed method with the use of surface water balance principle has a more solid physical basis
than previous downscaling methods. By introducing SSM as an auxiliary variable, the impact of inherent bias in satellite
estimates on the downscaled results can be moderately reduced compared to the conventional statistical method. The
validation with rain gauge data highlights the importance of SSM as a fully independent source of information that can

be effectively used for downscaling coarse-resolution precipitation at a daily scale, which is rarely conducted in current related studies. Therefore, this method is a promising way to derive high-resolution precipitation data and shows good potential for real-time precipitation data downscaling with the provision of SSM data, which will assist further applications in related fields (such as hydrology, agriculture, natural hazards, water resources, and climate change).

**Code and data availability**

This study used the surface soil moisture data with high resolution (https://doi.org/10.5281/zenodo.7451422) to produce the downscaled precipitation data (https://doi.org/10.5281/zenodo.7451690), which were available at the zenodo data survey portal. The part of observed data obtained on (https://www.ncei.noaa.gov/access/search/data-search/global-summary-of-the-day). The Matlab codes can be obtained upon request from the corresponding author.

**Author contributions**

Kunlong He led the investigation, conceptualized the study, designed the formal analysis, and wrote the initial draft. Wei Zhao was responsible for conceptualizing the study, investigating methods, obtaining the funding, supervising the study process, and reviewing and editing the paper. Luca Brocca conceptualized the research, reviewed the manuscript and provided the in-situ measurements. Pere Quintana-Seguí helped with the investigation, provided the datasets, and reviewed the paper.

**Declaration of Competing Interest**

The authors declare that they have no known competing financial interests or personal relationships that could have appeared to influence the work reported in this paper.

**Acknowledgments**

This research was partially funded as part of the National Natural Science Foundation of China (Grant No. 42071349), Sichuan Science and Technology Program (Grant No. 2020JDJQ0003), the West Light Foundation of the Chinese Academy of Sciences, and the project PRIMA PCI2020-112043 funded by MCIN/AEI/10.13039/501100011033. We thank the Spanish State Meteorological Agency (AEMET) for sharing daily precipitation data with this project.

**Review statement**

The authors sincerely thank Editor Shraddhanand Shukla and three anonymous referees for their insightful comments.

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
