# Peer review of "SMPD: A soil moisture-based precipitation downscaling method for"

_Hydrology and Earth System Sciences, 2022_

## Referee Comment (RC1)

"SMPD: A soil moisture-based precipitation downscaling method for high-resolution daily satellite precipitation estimation" by He et al.

Overall comments:

This aims of this study are very clear and significant. Precipitation dataset with fine spatio-temporal resolutions are of importance for climatological and meteorological investigations and applications. The method seems to be novelty, while the accuracy of the results have not achieved significant improvements. Considering the high standards of *HESS*, there are still various aspects needing to be greatly improved.

Specific comments:

1. Title: what are the capitals, SMPD, standing for? Please label them.

2. Title: would it be possible for downscaling the IMERG at hourly or half-hourly scale? For this point, I would like to recommend you some references:

   The first comparisons of IMERG and the downscaled results based on IMERG in hydrological utility over the Ganjiang River basin, Water, 10 (1392), 1-15, DOI: 10.3390/w10101392.

   AERA5-Asia: A long-term Asian precipitation dataset (0.1°, 1 hourly, 1951–2015, Asia) anchoring the ERA5-Land under the total volume control by APHRODITE, Bulletin of American Meteorological Society, Bulletin of American Meteorological Society, 103 (4) DOI: 10.1175/BAMS-D-20-0328.1.

   AIMERG: a new Asian precipitation dataset (0.1°/half-hourly, 2000–2015) by calibrating GPM IMERG at daily scale using APHRODITE, Earth System Science Data, 12(3): 1525-1544, DOI: 10.5194/essd-2019-250.

Calibrating GPM IMERG Late-Run product using ground-based CPC daily precipitation data: a case study in the Beijing-Tianjin-Hebei urban agglomeration, Remote Sensing Letters, 12:9, 848-858. DOI: 10.1080/2150704X.2021.1942576.

3. Introduction: this part demonstrates that the authors have not comprehensively known the key spatial downscaling investigations, and the writings is really not good. Please reorganize and rewrite it.

A Spatial Data Mining Algorithm for Downscaling TMPA 3B43 V7 Data over the Qinghai-Tibet Plateau with the Effect of Systematic Anomalies Removed. Remote Sensing of Environment 200: 378-395. DOI: 10.1016/j.rse.2017.08.023

Respective Advantages of "Top-Down" Based GPM IMERG and "Bottom-Up" Based SM2RAIN-ASCAT Precipitation Products Over the Tibetan Plateau, Journal of Geophysical Research: Atmospheres, 126, e2020JD033946. DOI: https://doi.org/10.1029/2020JD033946.

Spatially Downscaling IMERG at Daily Scale using Machine Learning Approaches over Zhejiang, Southeastern China, Frontiers in Earth Science, 8:146. DOI: 10.3389/feart.2020.00146.

Long Term Precipitation Estimates Generated by a Downscaling Calibration Procedure Over the Tibetan Plateau From 1983 to 2015. Earth and Space Science, 6, 2180-2199. DOI: 10.1029/2019EA000657.

A new approach for obtaining precipitation estimates with finer spatial resolution at daily scale based on TMPA V7 data over the Tibetan Plateau, International Journal of Remote Sensing, 40:22, 8465-8483. DOI:10.1080/01431161.2019.1612118.

Downscaling Annual Precipitation with TMPA and Land Surface Characteristics in China. International Journal of Climatology 37: 5017-5119. DOI: 10.1002/joc.2017.37.issue-15.

4. Datasets-IMERG: Various investigations have been done to exploiting the potential errors in IMERG. If the authors point out some potential error sources of IMERG, it is much better.

   Do ERA5 and ERA5-Land Precipitation Estimates Outperform Satellite-based Precipitation Products? A Comprehensive Comparison between State-of-the-art Model-based and Satellite-based Precipitation Products over Mainland China. Journal of Hydrology, 605: 127353, DOI: 10.1016/j.jhydrol.2021.127353.

5. Datasets-SSM: how do you think that whether the too coarse spatial resolution of CCI SSM data, 0.25 deg, have negative efforts in downscaling IMERG at 0.1 deg or not?

6. Datasets-NDVI: how do you think that whether the too coarse temporal resolution of CCI SSM data, 16-day, have negative efforts in downscaling IMERG at daily scale or not?

7. Validation: why did not using the POD index, which is a very common index evaluating precipitation datasets.

   Quantitative Evaluations and Error Source Analysis of Fengyun-2-Based and GPM-Based Precipitation Products over Mainland China in Summer, 2018, Remote Sensing, 11(24):2992. DOI: 10.3390/rs11242992.

   Spatiotemporal Assessments on the Satellite‐Based Precipitation Products From Fengyun and GPM Over the Yunnan‐Kweichow Plateau, China. Earth and Space Science, 7, e2019EA000857. DOI: 10.1029/2019EA000857

8. Results: The idea of Fig.3 is not very clear.

9. Results: the downscaled results on 20171210 in the central part seems have anomalies, why?

10. Discussion: would it be possible to analyze the potential error sources of the downscaled results?

11. The English writings are also greatly needed to be improved.

12. Last but most important one: would you like to use some traditional method as a comparison with your proposed method, SMPD?

---

## Author Response (AR1)

Dear Dr. Shraddhanand Shukla:

Thank you very much for your great help on our manuscript. Many thanks are also to the Associate Editor and the reviewers for their constructive comments and suggestions. They are very important and useful to improve our work and brush up the manuscript. According to all the comments, the paper was thoroughly revised. Meanwhile, some errors and deficiencies have been also revised through our self-check process and proofread service. The key changes are marked with red color. The point-to-point responses to all comments and suggestions from the reviewer are listed in the following. We hope these revisions can satisfy your requirements and meet with your approval, and of course, we are more than happy to improve the paper again according to new comments and suggestions they might come.

Best regards,

Kunlong He and Wei Zhao
Corresponding author: Wei Zhao  Prof
E-mail: zhaow@imde.ac.cn

**Responses**

(1) How is the proposed method different and improved over a similar approach Ciabatta et al, 2018. That paper also uses ESA CCI soil moisture dataset.

**Ans.:** Thanks a lot for your good comment. This study has some similarities in the terms of the datasets and approach conducted by Ciabatta et al, 2018. The ESA CCI soil moisture product was used in both studies to inversion the rainfall estimation model based on the soil water balance equation. According to the estimation evaluation in Ciabatta et al, 2018, ESA CCI soil moisture has a big advantage in estimating precipitation and this connection was approved by the SM2RAIN-CCI method in this study. However, the purpose of Ciabatta et al. (2018) is to generate a new global long-term rainfall dataset without the improvement of spatial resolution. The estimated rainfall product keeps to the same resolution as the ESA CCI product with the value of 0.25 °. In contrast, this study focuses on spatially downscaling satellite precipitation products with a seamless high spatial resolution (1km) soil moisture product. Its aim is quite different from that of Ciabatta et al. (2018). This study is an extension of previous SM2RAIN method to the field of spatial disaggregation and the results confirmed the reliability of this approach in generating high-resolution precipitation data with acceptable accuracy. We have revised the manuscript to clarify the purpose of this study. 'Thus, the main objective of this study is to establish a soil moisture-based precipitation downscaling (SMPD) scheme as a novel way of obtaining fine-scale precipitation by fragmenting the coarse-pixel rainfall to fine-scale pixels'.

(2) What is the reason for focusing only on 2016-2018 data, both IMERG and ESA CCI are available for a much longer period?

**Ans.:** Thanks a lot for your comment. Indeed, both IMERG and ESA CCI are available for a much longer period. However, for the research period, it is based on the availability of the rain gauge data, which is assessable for the period from 2016 to 2018. We have clarified this point in the data section. Meanwhile, as a methodology study, we believe the analysis at this three-year period is appropriate to demonstrate the performance of the proposed method.

(3) What is the improvement in downscaling through the proposed approach relative to a simple interpolation based downscaling method?

**Ans.:** As you indicated, the interpolation method is indeed convenient and simple for coarse-resolution precipitation data downscaling. However, this type method was only conducted with some mathematical processes but poor of physical background. Consequently, it may have great uncertainty in the process of obtaining high-resolution precipitation data, especially in the region with complex terrain (Guo et al., 2021; Zhang et al., 2020). Table1 shows that the evaluation index of precipitation data generated by SMPD method is generally better than that generated by simple interpolation methods. The downscaling method proposed by us takes into account the physical process of precipitation and adopts a bottom-up model to downscale satellite precipitation products. We have added some discussions in the discussion section. See new lines from 470 to 474 of page 21.

Table 1 The evaluation indexes of original GPM precipitation product, downscaled GPM precipitation product, resampling GPM precipitation product based on bilinear interpolation method and gauge-based precipitation data based on bilinear interpolation method.

|  | Original | Downscaled | Bilinear-gpm | Bilinear-observation |
|---|---|---|---|---|
| CC | 0.60 | 0.61 | 0.58 | 0.40 |

| | | | | |
|---|---|---|---|---|
| RMSE | 4.99 | 4.83 | 4.87 | 5.06 |
| BIAS | 9% | 5% | 9% | -1% |
| POD | 0.84 | 0.88 | 0.78 | 0.67 |
| FAR | 0.52 | 0.47 | 0.47 | 0.49 |
| CSI | 0.44 | 0.48 | 0.45 | 0.40 |

(4) How would this approach work for real-time (i.e. IMERG-Early) rainfall data?

**Ans.:** Thanks for your comments. In the proposed method, the key inputs of the downscaling process is surface soil moisture and precipitation data. As introduced in the method section, the soil moisture estimation method has achieved seamless downscaling for high-resolution soil moisture generation under cloudy conditions. Therefore, it would be able to obtain real-time soil moisture from microwave satellite observations combined with surface temperature and vegetation index derived from optical and thermal infrared remote sensing. Therefore, this approach would work well for real-time rainfall data. We have clarified this point in the conclusion section. See new lines from 543 to 552 of page 25.

Reviewer #1:

(1). Title: what are the capitals, SMPD, standing for? Please label them.

**Ans.:** Thanks a lot for your comment. SMPD stands for Soil Moisture-based Precipitation Downscaling method. We labeled it in this paper. See new lines 14 and 201.

(2). Title: would it be possible for downscaling the IMERG at hourly or half-hourly scale? For this point, I would like to recommend you some references.

**Ans.:** Thanks a lot for your comment. In the proposed method, the key inputs of the downscaling process is surface soil moisture and precipitation data. Even on hourly or half-hourly scales, the soil moisture exhibits an instantaneous response to collocated precipitation. Then, as introduced in the method section, the soil moisture estimation method has achieved seamless downscaling for high resolution soil moisture generation under cloudy conditions. Therefore, it would be able to obtain real-time soil moisture from microwave satellite observations combined with surface temperature and vegetation index derived from optical and thermal infrared remote sensing. Therefore, this approach would work well for rainfall data at hourly or half-hourly scale. We will be looking at this in future studies and adding this part to the discussion (see new lines from 527 to 537 of section 5.2). Thank you very much for the recommended references, which we have read and referenced in this paper.

(3). Introduction: this part demonstrates that the authors have not comprehensively known the key spatial downscaling investigations, and the writings is really not good. Please reorganize and rewrite it.

**Ans.:** Thanks a lot for your comment. We have read the literature you provided and added more methodological descriptions in the introduction section, and our research points are reorganized and rewrote 'Furthermore, suffering from an indirect physical connection between topographic and vegetation factors and precipitation at coarse temporal scale, a large amount of downscaling research has been conducted at monthly or annual scales. In addition, although daily high-resolution precipitation data have been produced by different statistical methods (Brocca et al., 2019a; Hong et al., 2021), the use of high-resolution SSM data to improve the spatial resolution of satellite precipitation products for generating daily-scale high-resolution precipitation data based

on physical mechanisms is less studied. See new lines from 90 to 93 of page 4.

(4). Datasets-IMERG: Various investigations have been done to exploiting the potential errors in IMERG. If the authors point out some potential error sources of IMERG, it is much better.

**Ans.:** Thanks a lot for your comment. We added the potential error sources analysis of IMERG in the discussion section of the paper. 'At first, the IMERG-Final products are corrected on a monthly scale using the interpolated precipitation product Global Precipitation Climatology Centre (GPCC, 1.0°/Monthly) based on ground observations. However, there is no mature calibration algorithm for calibrating the daily satellite-based precipitation estimates (Ma et al., 2020b). Second, the a-priori databases of cloud cover and precipitation profiles for retrieving passive microwave-based satellite precipitation estimates are not sufficiently robust due to the lack of ground-based radar observations. In addition, since passive microwave remote sensing-based precipitation retrieval is the primary input to the IMERG-Final products, it may lead to poor performance of the satellite-based product in winter and high-latitude regions (Xu et al., 2022)'. Please see new lines from 511 to 520 of page 24.

(5). Datasets-SSM: how do you think that whether the too coarse spatial resolution of CCI SSM data, 0.25 deg, have negative efforts in downscaling IMERG at 0.1 deg or not?

**Ans.:** Thanks a lot for your comment. The spatial resolution of surface soil moisture data of 25 km is very coarse and cannot meet the precipitation downscaling, while the surface soil moisture data are easily affected by clouds. Therefore, in this study, the 25-km European Space Agency (ESA) Climate Change Initiative (CCI) SSM product is used to derive 1-km SSM data based on the seamless downscaling method proposed by Zhao et al. (2021). Therefore, the 1 km precipitation data can mitigate the impact of the coarse resolution of GPM precipitation products. In addition, the uncertainty of SSM and the sensitivity relationship between SSM and precipitation under continuous rainfall conditions may introduce uncertainty in the downscaling precipitation results. First, the responses of SSM with different land cover conditions and vegetation coverages to precipitation are relatively different (Fan et al., 2021), and topographic factors such as depressions and slopes also affect the uncertainty of SSM. Second, although the relationship between SSM and precipitation has been well demonstrated in many previous studies (Brocca et al., 2016; Crow et al., 2009), the sensitivity of SSM to precipitation will decrease when soil water storage becomes saturated after repeated precipitation. Therefore, it is necessary to further improve the relationship by considering the soil water threshold saturation in future studies. We investigated this limitation in the discussion section of the paper. Please see new lines from 527 to 537 of page 24.

(6). Datasets-NDVI: how do you think that whether the too coarse temporal resolution of CCI SSM data, 16-day, have negative efforts in downscaling IMERG at daily scale or not?

**Ans.:** Thanks a lot for your comment. Some studies stated that the precipitation–NDVI relationship was hardly time-delayed since vegetation could influence precipitation by adjusting temperature and air moisture during the growing seasons (Chen et al., 2020; Lu et al., 2022). The NDVI values within the 16-day scale remain largely constant and respond well to precipitation, while vegetation dissipates precipitation through leaf interception and evapotranspiration. Thus, we believe that the 16-day NDVI does not negatively affect the downscaling results at daily scale.

(7). Why did not using the POD index, which is a very common index evaluating precipitation datasets.

**Ans.:** Thanks a lot for your comment. We added the POD index results in the results section of the paper. Please see Fig.6, Fig.7 (j, k, and l), and Table. 1 and 2 in the manuscript.

(8). Results: The idea of Fig.3 is not very clear.

**Ans.:** Thanks a lot for your comment. We proposed the soil moisture-based precipitation estimation model based on equation 6. An important prerequisite was the assumption of spatial invariancy in the precipitation estimation model described in Eq. (6) at coarse and fine scales. Thus, we compared the precipitation results fitted by equation 7 with the original GPM products over the study area during the period of 2016-2018, the evaluation indicators including correlation coefficients (CC) root mean square error (RMSE) and corresponding standard deviation were calculated (Fig . 3), which could evaluate the performance of the soil moisture-based precipitation estimation model. Please see new lines from 279 to 285 of page 11.

(9). Results: the downscaled results on 20171210 in the central part seems have anomalies, why?

**Ans.:** Thanks a lot for your comment. These anomalies may be due to the uncertainty of SSM and the sensitivity relationship between SSM and precipitation under continuous rainfall conditions may introduce uncertainty in the downscaling precipitation results. First, the responses of SSM with different land cover conditions and vegetation coverages to precipitation are relatively different, and topographic factors such as depressions and slopes also affect the uncertainty of SSM. Second, although the relationship between SSM and precipitation has been well demonstrated in many previous studies (Brocca et al., 2014; Brocca et al., 2016; Mao et al., 2018), the sensitivity of SSM to precipitation will decrease when soil water storage becomes saturated after repeated precipitation. Therefore, we re-changed the code in these regions, using either spatial proximity image element or temporal proximity image element values to fill the outliers (see Fig. 5).

(10). Discussion: would it be possible to analyze the potential error sources of the downscaled results?

**Ans.:** Thanks a lot for your comment. We added the potential error sources of the downscaled results in section 5.2. Please see new lines from 511 to 520 of page 24. 'In addition, the uncertainty of SSM and the sensitivity relationship between SSM and precipitation under continuous rainfall conditions may introduce uncertainty in the precipitation downscaling results. First, the responses of SSM with different land cover conditions and vegetation coverages to precipitation are relatively different (Fan et al., 2021), and topographic factors such as depressions and slopes also affect the uncertainty of SSM. Therefore, it is necessary to establish the relationship between SSM and precipitation for different land cover types or different terrain types. The establishment of a more reliable fitting relationship based on precipitation data with different land cover properties or topographic factors would be helpful to enhance the accuracy of the downscaling results (Chen et al., 2020; Senanayake et al., 2021; Zhao et al., 2021). Second, although the relationship between SSM and precipitation has been well demonstrated in many previous studies (Brocca et al., 2014; Brocca et al., 2016; Mao et al., 2018), the sensitivity of SSM to precipitation will decrease when soil water storage becomes saturated after repeated precipitation. Therefore, it is necessary to further improve the relationship by considering the soil water threshold saturation in future studies.'

(11). The English writings are also greatly needed to be improved.

**Ans.:** Thanks a lot for your comment. We have carefully checked the language and rewritten some parts of the manuscript.

(12). Last but most important one: would you like to use some traditional method as a comparison with your proposed method, SMPD?

**Ans.:** Thanks a lot for your comment. We collected the downscaled results evaluation metrics of existing studies on GPM precipitation products in Table 3 and indicated the superiority of SMPD method by comparing with various methods (e, g. GWR). Additionally, the comparison by bilinear interpolation method to interpolate the original GPM precipitation products to 0.01 resolution reveals that the SMPD method has a slight improvement compared to the interpolation method. The accuracy of interpolation precipitation based on rain gauge stations is limited by the density of gauge-based stations, and the GWR method is limited by the model window radius and the influence of the number of gauge-based stations. The SMPD method breaks the limitation caused by the rainfall gauge density and the model window radius, which has a promising application prospect to generate precipitation data with high resolution and high accuracy in the study area with heterogeneous terrain morphology and precipitation. Please see lines from 505 to 506 of page 23.

Reviewer #2:

(1). About the spatial distribution of the parameters of a, b, and c. These three parameters are used in equations 7 and 8. According to my understanding, they were calibrated at 10 km resolution and then applied to 1 km resolution. So, firstly, are these parameters scale independent? Moreover, are they also temporal-independent?

**Ans.:** Thanks a lot for your comment. To perform the precipitation downscaling, an important prerequisite is the assumption of spatial invariancy in the precipitation estimation model described in Eq. (6) at coarse and fine scales, which is also the basis of many related downscaling studies for other parameters, such as surface soil moisture and temperature (Hutengs and Vohland, 2016; Mishra et al., 2018; Zhao et al., 2018; Ebrahimy and Azadbakht, 2019). Therefore, these parameters are not scale-independent, the estimated parameters at 10 km resolution scale can be used for the corresponding 100 sub-images units (1 km). Moreover, because we construct models using self-adaptive windows in different local regions on the daily scale, these parameters vary in time and space. Thus, they are also temporal independent. We have clarified the above two points in the revised manuscript. See new lines from 206 to 209 of page 8.

(2). The accuracy of the high-resolution SSM data. The authors are suggested to show the reliability of this new information before downscaling.

**Ans.:** Thanks a lot for your comment. In fact, the accuracy of the downscaled precipitation results depends on seamless high-resolution soil moisture data. Therefore, SSM is an important variable in the estimation model. The ESA CCI SSM product can only provide coarse-resolution SSM data with unexpected gaps. To obtain daily SSM at a 1-km resolution, the seamless SSM downscaling method proposed by Zhao et al. (2021) is a good choice to achieve this goal. The proposed method was successfully applied to data obtained for the Iberian Peninsula from January 1, 2016 to December 31, 2018. Based on the comparison with the precipitation dataset, the downscaled SSM exhibited strong temporal correlation with rainfall events. Evaluation using the in situ SSM from the REMEDHUS network highlighted the good performance of the downscaled SSM at network level with a correlation coefficient (R) of 0.820. The root-mean-square-error (RMSE), unbiased root-mean-square error ($ub$RMSE), and bias were 0.091, 0.033, and 0.085 $m^3/m^3$, respectively (as shown in table 2). These results confirmed that the proposed method is an efficient and convenient downscaling process that can be successfully used to generate high-resolution SSM data without spatiotemporal gaps. Therefore, based on the seamless, high spatial resolution and high accuracy soil moisture data produced by this method, we believe that it could meet the accuracy requirements of the downscaling process for precipitation. In the revised manuscript, we added explanations to show the reliability of the high-resolution SSM for the precipitation downscaling. See new lines from 215 to 218 of page 8.

Table 2

Comparison of the statistical metrics of the ESA CCI SSM and downscaled SSM with the *in situ* SSM at station level on days with ESA CCI SSM values in the study area (N represents the sample number, **significant at the 99% confidence level, bold numbers indicate the best scores, and *italic* numbers indicate the worst scores).

| Station ID | Downscaled SSM against *in-situ* SSM | | | | | ESA CCI SSM against *in-situ* SSM | | | | |
|---|---|---|---|---|---|---|---|---|---|---|
| | R | RMSE($m^3/m^3$) | *ub*RMSE($m^3/m^3$) | Bias($m^3/m^3$) | N | R | RMSE($m^3/m^3$) | *ub*RMSE($m^3/m^3$) | Bias($m^3/m^3$) | N |
| K13 | *0.459*** | 0.098 | *0.084* | −0.050 | 867 | *0.456*** | 0.093 | *0.083* | −0.041 | 811 |
| K10 | 0.673** | 0.159 | 0.044 | 0.153 | 898 | 0.697** | 0.166 | 0.041 | 0.160 | 848 |
| M5 | 0.814** | 0.101 | 0.034 | 0.095 | 898 | 0.817** | 0.100 | 0.033 | 0.095 | 848 |
| N9 | 0.731** | 0.056 | 0.049 | 0.028 | 897 | 0.713** | 0.059 | 0.050 | 0.032 | 818 |
| I6 | 0.708** | 0.192 | 0.048 | 0.186 | 886 | 0.691** | 0.190 | 0.046 | 0.185 | 836 |
| H7 | 0.694** | *0.199* | 0.048 | *0.193* | 873 | 0.720** | *0.198* | 0.044 | *0.193* | 826 |
| K9 | 0.715** | 0.130 | 0.049 | 0.121 | 868 | 0.737** | 0.137 | 0.048 | 0.128 | 825 |
| H9 | 0.673** | 0.087 | 0.077 | −0.042 | 825 | 0.662** | 0.084 | 0.077 | −0.033 | 777 |
| J14 | 0.816** | 0.106 | 0.045 | 0.096 | 568 | 0.838** | 0.107 | 0.044 | 0.098 | 524 |
| M9 | 0.749** | 0.063 | 0.043 | 0.046 | 898 | 0.726** | 0.069 | 0.044 | 0.053 | 848 |
| F6 | 0.783** | 0.056 | 0.050 | 0.026 | 892 | 0.810** | 0.058 | 0.047 | 0.034 | 865 |
| H13 | 0.846** | 0.071 | 0.031 | 0.064 | 869 | 0.852** | 0.062 | 0.030 | 0.054 | 819 |
| L3 | 0.596** | 0.133 | 0.048 | 0.124 | 873 | 0.589** | 0.131 | 0.047 | 0.122 | 823 |
| J12 | 0.802** | 0.073 | 0.045 | −0.058 | 896 | 0.798** | 0.072 | 0.045 | −0.057 | 838 |
| E10 | 0.650** | 0.118 | 0.057 | 0.104 | 895 | 0.674** | 0.116 | 0.055 | 0.102 | 868 |
| O7 | 0.824** | 0.139 | 0.034 | 0.135 | 881 | 0.807** | 0.142 | 0.033 | 0.138 | 803 |
| K4 | 0.709** | *0.199* | 0.044 | *0.193* | 894 | 0.701** | 0.195 | 0.043 | 0.190 | 844 |
| L7 | 0.744** | 0.053 | 0.046 | 0.027 | 898 | 0.781** | 0.053 | 0.042 | 0.032 | 848 |
| J3 | 0.771** | 0.197 | 0.042 | 0.192 | 898 | 0.810** | 0.194 | 0.037 | 0.190 | 848 |
| F11 | 0.912** | 0.139 | 0.026 | 0.136 | 896 | 0.897** | 0.139 | 0.026 | 0.137 | 790 |
| Average | 0.733 | 0.118 | 0.047 | 0.088 | 868.5 | 0.739 | 0.118 | 0.046 | 0.091 | 815.35 |

(3). Validation of precipitation data. Currently, the authors are using pixel-point matching comparison. It is suggested to upscale the point observation of stations to grid scales of 1 km and 10 km. And then comparing the two precipitation data to corresponding ground observation.

**Ans.:** Very good comment. To date, most studies used rain gauge stations to evaluate the performance of satellite precipitation products and downscaled products because the gauge-based observations are taken as the most accurate precipitation values Using mathematical interpolation method (e.g., Kriging, IDW) to upscale the point observation to grid scales of 10 km and 1 km scales is an effective tool, but these methods may introduce large uncertainties in the upscaled results and lead to poor performance in evaluating the downscaled results (Xiaojun et al., 2021; Zhang et al., 2020). In addition, because the performance of the upscaled results depends on the gauge-based stations density, we will use the upscaled results of rain gauge stations to evaluate the downscaled results in the area of high-density gauge-based stations in future studies (Abdollahipour et al., 2021; Chen et al., 2021; Chena et al., 2018; Ma et al., 2017). About the spatial inconsistency of the point measurement and grid-scale estimation, we added discussions in the manuscript and it should be paid attention in future comparison. See new lines from 523 to 526 of page 24.

(4) As my speculation, there are more heavy rainfall (big rain rate values) events in high resolution precipitation data. However, it is not shown in the histogram of figures 4 and 7. The authors are suggested to check this issue with rain gauge observations.

**Ans.:** Thanks a lot for your comment. We have added explanation of this point in the revised manuscript as below: "To assess the GPM products' performance at different precipitation intensity events. The daily precipitation intensity is classified into five categories, and the rainfall thresholds are classified as 0, 10, 20, 40 mm respectively. The performance metrics for the five daily precipitation intensity classes listed in Table 3. In summary, the original and downscaled GPM products performed the best in terms of all performance metrics for the no-rain events, while performed the worst for the violent rain events ($> 40$ mm d$^{-1}$). All precipitation products indicated that FAR values continuously performed the worst for the violent rain intensities, which showed that the products are still unable to accurately capture high precipitation values. Due to the reduced FAR values, the CSI value performed the best for no-rain events, followed by light rain ([0, 10] mm d$-1$), moderate rain ([10, 20] mm d$^{-1}$), heavy rain, ([20, 40] mm d$^{-1}$) and violent rain, respectively. Additionally, the BIAS values showed that all precipitation products overestimated

the number of light rain and underestimated moderate rains, heavy rains, and violent rains. Most importantly, the accuracy of the downscaled product was slightly better than the original precipitation product for different rainfall intensity events in terms of CC, RMSE, POD, FAR and CSI values, indicating the reliability of the downscaled products in capturing different rainfall intensity events. " see new lines from 409 to 423 of pages 18 and 19.

Table 3. CC, RMSE, BIAS, POD, FAR and CSI values for the different precipitation intensities for original and downscaled GPM products from 2016 to 2018.

| Intensity (mm) | Original | | | | | | Downscaled | | | | | |
|---|---|---|---|---|---|---|---|---|---|---|---|---|
| | CC | RMSE (mm) | BIAS (%) | POD | FAR | CSI | CC | RMSE (mm) | BIAS (%) | POD | FAR | CSI |
| 0 | - | 1.83 | - | 0.93 | 0.34 | 0.63 | - | 1.73 | - | 0.94 | 0.26 | 0.70 |
| 0-10 | 0.30 | 6.39 | 27.00 | 0.69 | 0.65 | 0.31 | 0.30 | 5.98 | 23.00 | 0.73 | 0.60 | 0.34 |
| 10-20 | 0.15 | 11.85 | -20.00 | 0.26 | 0.75 | 0.15 | 0.15 | 11.50 | -22.00 | 0.25 | 0.74 | 0.15 |
| 20-40 | 0.15 | 18.41 | -33.00 | 0.25 | 0.78 | 0.13 | 0.14 | 18.31 | -36.00 | 0.26 | 0.77 | 0.14 |
| >40 | 0.28 | 39.53 | -47.00 | 0.23 | 0.84 | 0.11 | 0.28 | 39.33 | -50.00 | 0.25 | 0.82 | 0.12 |

References

Abdollahipour, A., Ahmadi, H., Aminnejad, B., 2021. A review of downscaling methods of satellite-based precipitation estimates. Earth Science Informatics: 1-20.

Brocca, L., Ciabatta, L., Massari, C., Moramarco, T., Hahn, S., Hasenauer, S., Kidd, R., Dorigo, W., Wagner, W., Levizzani, V., 2014. Soil as a natural rain gauge: Estimating global rainfall from satellite soil moisture data. J. Geophys. Res. - Atmos., 119(9): 5128-5141. DOI:https://doi.org/10.1002/2014JD021489

Brocca, L., Pellarin, T., Crow, W.T., Ciabatta, L., Massari, C., Ryu, D., Su, C.H., Rüdiger, C., Kerr, Y., 2016. Rainfall estimation by inverting SMOS soil moisture estimates: A comparison of different methods over Australia. Journal of Geophysical Research Atmospheres, 121(20): 12,062-12,079.

Chen, C., Hu, B., Li, Y., 2021. Easy-to-use spatial random-forest-based downscaling-calibration method for producing precipitation data with high resolution and high accuracy. Hydrol. Earth Syst. Sci., 25(11): 5667-5682. DOI:10.5194/hess-25-5667-2021

Chena, Y., Huanga, J., Shengd, S., Mansaraya, L.R., Wangh, X., 2018. A new downscaling-integration framework for high-resolution monthly precipitation estimates: Combining rain gauge observations, satellite-derived precipitation data and geographical ancillary data. Remote Sensing of Environment, 214: 154-172.

Crow, W.T., Huffman, G.J., Bindlish, R., Jackson, T.J., 2009. Improving satellite-based rainfall accumulation estimates using spaceborne surface soil moisture retrievals. Journal of Hydrometeorology, 10(1): 199-212.

Guo, X., Guo, Cui, P., Chen, X., Li, Y., Zhang, J., Sun, Y., 2021. Spatial uncertainty of rainfall and its impact on hydrological hazard forecasting in a small semiarid mountainous watershed. J. Hydrol., 595: 126049. DOI:https://doi.org/10.1016/j.jhydrol.2021.126049

Ma, Z., Shi, Z., Zhou, Y., Xu, J., Yu, W., Yang, Y., 2017. A spatial data mining algorithm for

downscaling TMPA 3B43 V7 data over the Qinghai–Tibet Plateau with the effects of systematic anomalies removed. Remote Sensing of Environment, 200: 378-395.

Mao, Y., Crow, W., Nijssen, B., 2018. A Framework for Diagnosing Factors Degrading the Streamflow Performance of a Soil Moisture Data Assimilation System. Journal of Hydrometeorology, 20. DOI:10.1175/JHM-D-18-0115.1

Xiaojun, G., Peng, C., Xingchang, C., Yong, L., Ju, Z., Yuqing, S., 2021. Spatial uncertainty of rainfall and its impact on hydrological hazard forecasting in a small semiarid mountainous watershed. Journal of Hydrology, 595: 126049.

Zhang, L., Ren, D., Nan, Z., Wang, W., Zhao, Y., Zhao, Y., Ma, Q., Wu, X., 2020. Interpolated or satellite-based precipitation? Implications for hydrological modeling in a meso-scale mountainous watershed on the Qinghai-Tibet Plateau. Journal of Hydrology, 583: 124629. DOI:https://doi.org/10.1016/j.jhydrol.2020.124629

Zhao, W., Wen, F., Wang, Q., Sanchez, N., Piles, M., 2021. Seamless downscaling of the ESA CCI soil moisture data at the daily scale with MODIS land products. J. Hydrol., 603: 126930. DOI:https://doi.org/10.1016/j.jhydrol.2021.126930

---

## Referee Report (RR1)

In this manuscript, authors developed a precipitation downscaling method based on a fine soil moisture product. It's an interesting topic to improve the spatial details of satellite-based precipitation. My comments are as follows.

1. As shown in figure 1, changes of soil moisture  $\left(\frac{ds(t)}{dt}\right)$  significantly lagged behind precipitation events, but the authors didn't consider this scenario.

2. Methodology: to facilitate the description, I suggests adding an algorithm flowchart.

3. Is the air temperature  $(T_a)$  in eq. 4 time varying? The air temperature can be assumed to be the same in a small extent, but the temporal variation in temperature is inevitable and cannot be ignored.

4. The description of the residual correction in 3.23 (particularly for eq.9 and eq.10) is rather ambiguous and please check eq.9 again.

5. In section 3.3, the *i* in eq. 12 - 14 stands for station but in figure 7 the *i* seems stands for the sampling time/number.

6. Section 4.1: Does the soil moisture-based precipitation estimation model used in section 4.1 contain the residual correction?

7. Line 296 – 303: This paragraph can be combined in the section of method.

8. Please provide additional explanation to the meaning of the points shown in Figures 6 and 9, and provide a description of how to obtain the statistical metric (also for table 1 and table 2) since they contain both temporal and spatial information.

---

## Author Response (AR2)

Dear Editor and Reviewers:

Thank you very much for your review on our manuscript. Your constructive comments and suggestions are very important and useful to improve our work and brush up the manuscript.

According to all the comments, the paper was thoroughly revised with point-to-point response. Meanwhile, some errors and deficiencies have been also revised through our self-check process and proofread service. The key changes are marked with red color. The detailed responses are listed in the following. We hope these revisions can satisfy your requirements and meet with your approval for final publication in HESS.

Best regards,

Wei Zhao and on behalf of all co-authors Institute of Mountain Hazards and Environment, CAS E-mail: zhaow@imde.ac.cn

**Reply to Referee #3**

In this manuscript, authors developed a precipitation downscaling method based on a fine soil moisture product. It's an interesting topic to improve the spatial details of satellite-based precipitation. My comments are as follows:

(1) As shown in figure 1, changes of soil moisture  $\frac{ds(t)}{dt}$  significantly lagged behind precipitation events, but the authors didn't consider this scenario.

**Ans.:** Thanks a lot for your good comment. As a common knowledge, the difference between soil moisture observations at current and previous time steps can be used to estimate precipitation at current time. However, it is an important issue in this estimation that the changes of soil moisture at different depth lag behind precipitation events with different extents. According to the artificial rainfall-runoff experiment of Song et al. (2020), the results showed that this relationship can be improved by increasing the temporal aggregation interval. The performance improvement with the increase of aggregation intervals is in line with earlier findings at daily scale (Brocca et al. 2014; Brocca et al. 2016). Thus, the lagged response of soil moisture to precipitation can be ignored in the downscaling process of this study. We clarified this point in the revised version.

"Although there is a lagging effect of the changes in soil moisture to precipitation, the rainfall runoff experiment conducted by Song et al. (2020) further confirmed this effect becomes small with the increase of the temporal aggregation interval and its impact is relatively small at daily time scale (Brocca et al., 2016b)"

**Refs:**

- Brocca, L., Ciabatta, L., Massari, C., Moramarco, T., Hahn, S., Hasenauer, S., Kidd, R., Dorigo, W.,
  Wagner, W., & Levizzani, V. (2014). Soil as a natural rain gauge: Estimating global rainfall from satellite soil moisture data. *Journal of Geophysical Research: Atmospheres, 119*, 5128-5141
- Brocca, L., Pellarin, T., Crow, W.T., Ciabatta, L., Massari, C., Ryu, D., Su, C.H., Rüdiger, C., & Kerr, Y. (2016). Rainfall estimation by inverting SMOS soil moisture estimates: A comparison of different methods over Australia. *Journal of Geophysical Research: Atmospheres*, 121, 12,062-012,079
- Song, S., Brocca, L., Wang, W., & Cui, W. (2020). Testing the potential of soil moisture observations to estimate rainfall in a soil tank experiment. *Journal of Hydrology*, *581*, 124368

(2) Methodology: to facilitate the description, I suggest adding an algorithm flowchart.Ans.: Thanks a lot for your suggestion. We have added an algorithm flowchart in the methodology section, shown as follows:

(3) Is the air temperature (Ta) in eq. 4 time varying? The air temperature can be assumed to be the same in a small extent, but the temporal variation in temperature is inevitable and cannot be ignored.

**Ans.:** Thanks a lot for your comment. In *Eq.* 4, we consider that the  $T_a$  of each day is the same in a small extent. Thus, the term with the second brackets of Eq. 4 is simplified to a coefficient *c* (Eq. 2 as follows). Meanwhile, the coefficient *c* value is time-dependent, showing similar temporal variation feature as  $T_a$ .

$$e(t) = a \left( 1 - e^{-bVI} \right) \left( m / \left( 1 + e^{-(T_a - d)/p} \right) + f \right)$$
(1)

$$C = a \left( m / (1 + e^{-(T_a - d)/p}) + f \right)$$
(2)

We have clarified this process in the revised version.

(4) The description of the residual correction in 3.2.3 (particularly for eq. 9 and eq. 10 is rather ambiguous and please check eq. 9 again.

**Ans.:** Thanks for your comments. In the residual correction part, it is divided into two steps. First, the coarse resolution residual is interpolated by kriging interpolation method to obtain the high-resolution residual. Then, to meet the requirement of value preservation in the downscaling process, the kriging residuals should be corrected by redistributing it to each fine-resolution pixel *i*. That is, the ratio of the *i*th high-resolution residual pixel in the *j*th coarse-resolution cell to the sum of the precipitation in the *j*th coarse pixel is used as the weight, and the residual is multiplied by

the  $\lambda_{ij}$ , the kriging residuals were redistributed to each fine resolution pixel *i* to obtain the residual after value preservation. We have checked and modified this part, shown in section 3.2.3.

(5) In section 3.3, the *i* in eq. 12 – 14 stands for station but in figure 7 the *i* seems stands for the sampling time/number.

**Ans.:** Thanks for your comments. We modified this expression, *i* in *eq*. 12 - 14 stands for  $i^{th}$  time scale. In figure 7, all these indicators are calculated based on the comparison between the original precipitation estimates and downscaled precipitation estimates for all gauged-based stations from

2016 to 2018 periods, while *i* represents the  $i^{th}$  daily time scale, and the frequency statistics plot represents the frequency statistics of the index values for all evaluated sites rain gauge stations. We have added explanations to these terms in the revised manuscript.

**(6) Section 4.1: Does the soil moisture-based precipitation estimation model used in section 4.1 contain the residual correction?**

**Ans.:** The soil moisture-based precipitation estimation model used in section 4.1 does not contain the residual correction. First, we analyzed the soil moisture-based precipitation estimation model derivation process in Section 3.1 Then, Before the downscaling process, to evaluate the performance of the soil moisture-based precipitation estimation model based on the calibrated estimation model in Eq. 7. The mean value of the daily CCs and RMSEs during the period of 2016–2018 and their standard deviation (STD) by comparing the precipitation estimated with the proposed estimation model and the original GPM precipitation product was shown in Fig.4.

**(7) Line 296 - 303: This paragraph can be combined in the section of method.**

**Ans.:** Thanks for your suggestion. We are aware that this is the content of the downscaling method section, which has been combined with the methodology section 3.2.

(8) Please provide additional explanation to the meaning of the points shown in Figures 6 and 9, and provide a description of how to obtain the statistical metric (also for table 1 and table 2), since they contain both temporal and spatial information.

**Ans.:** Thanks for your comments. According to the formulas in section 3.3, Figures 6 and 9 (Figure 7 and 10 in the revised manuscript) sequentially show the scatter density plot of the original precipitation product and the downscaled precipitation data against in-situ observations from 1027 stations with the evaluation statistics (POD, FAR, CSI, R, RMSE and BIAS), at the daily and monthly scales during the period from 2016 to 2018.

Table 1 is the validation of the downscaled precipitation data, original GPM precipitation data with the daily precipitation measured by the selected stations at each month from 2016 to 2018. Table 2 represents the validation results at different precipitation density intervals for original and downscaled GPM products from 2016 to 2018 at daily scale. We provided additional explanation and description of the meaning of the figures and tables.